# Motor dysfunction and neurodegeneration in a *C9orf72* mouse line expressing poly-PR

Zongbing Hao[1], Liu Liu[1], Zhouteng Tao[2], Rui Wang[1], Haigang Ren[1], Hongyang Sun[1], Zixuan Lin[1], Zhixiong Zhang[1], Chenchen Mu[1], Jiawei Zhou [3] & Guanghui Wang [1]

A GGGGCC hexanucleotide repeat expansion in intron 1 of *chromosome 9 open reading frame 72* (*C9ORF72*) gene is the most common genetic cause of amyotrophic lateral sclerosis (ALS) and frontotemporal dementia. Repeat-associated non-ATG translation of dipeptide repeat proteins (DPRs) contributes to the neuropathological features of c9FTD/ALS. Among the five DPRs, arginine-rich poly-PR are reported to be the most toxic. Here, we generate a transgenic mouse line that expresses poly-PR (GFP-PR$_{28}$) specifically in neurons. GFP-PR$_{28}$ homozygous mice show decreased survival time, while the heterozygous mice show motor imbalance, decreased brain weight, loss of Purkinje cells and lower motor neurons, and inflammation in the cerebellum and spinal cord. Transcriptional analysis shows that in the cerebellum, GFP-PR$_{28}$ heterozygous mice show differential expression of genes related to synaptic transmission. Our findings show that GFP-PR$_{28}$ transgenic mice partly model neuropathological features of c9FTD/ALS, and show a role for poly-PR in neurodegeneration.

[1] Laboratory of Molecular Neuropathology, Jiangsu Key Laboratory of Neuropsychiatric Diseases & Department of Pharmacology, College of Pharmaceutical Sciences, Soochow University, Suzhou, Jiangsu 215123, China. [2] Center for Drug Safety Evaluation and Research, State Key Laboratory of Drug Research, Shanghai Institute of Materia Medica, Chinese Academy of Sciences, Shanghai 201203, China. [3] Institute of Neuroscience, State Key Laboratory of Neuroscience, Chinese Academy of Sciences Center for Excellence in Brain Science, Shanghai Institutes for Biological Sciences, Chinese Academy of Sciences, Shanghai 200031, China. Correspondence and requests for materials should be addressed to G.W. (email: wanggh@suda.edu.cn)

Amyotrophic lateral sclerosis (ALS) is a devastating neurodegenerative disease characterized by defects in upper and lower motor neurons. Mostly, the patients died of failure of the respiratory muscles within 1 to 5 years of disease onset[1,2]. Frontotemporal dementia (FTD) is a group of neurodegenerative disease characterized by progressive defects in frontal and temporal cortices[3]. Recent studies in ALS and FTD reveal that both diseases share many common genetic mutations and pathological features. A GGGGCC hexanucleotide-repeat expansion in C9ORF72 gene is the most common genetic cause of ALS and FTD[4–6].

The proposed pathological mechanisms of C9ORF72 gene mutation can be classified into three prototypes. First, loss of C9ORF72 protein function associates with cell death. Although C9ORF72 functions in endosome mature[7] and lysosome autophagy[8–10], C9orf72 null mice do not show motor neuron degeneration[11–14]. Second, RNA foci gain-of-function links to the pathology of c9FTD/ALS. The sense and antisense RNA foci have been found to sequester RNA-binding proteins in induced pluripotent stem cell (iPSC)-differentiated neurons[15–17], suggesting a gain of RNA toxicity. The BAC transgenic mouse lines display RNA foci and dipeptide-repeat protein (DPR) inclusions of c9FTD/ALS, either with or without a phenotype or neurodegeneration that may be associated with expression level, age, and repeat length in animals[12,18–20]. Third, gain of function by repeat-associated non-ATG initiated translation of five DPRs (poly-GA, GR, GP, PA, and PR) leads to DPR toxicity to neurons[21–27]. Notably, a recent publication demonstrated that C9ORF72 insufficiency causes neuron deficiency in coordination with toxic-repeat peptides[7], suggesting an involvement of multiple mechanisms.

Among the five DPRs, arginine-rich GR and PR DPRs have been reported to be highly toxic. Expression of GR or PR causes significant cell death in vitro[23,24]. In addition, the specific expression of GR or PR in Drosophila eyes causes severe degeneration[23,25,27,28]. What is more, several groups have generated C9ORF72 BAC transgenic mice, which suggest that gain of function may be a primary cause of c9FTD/ALS[12,18–20]. In GFP-GA$_{50}$-expressing mice, the mice developed poly-GA inclusions and behavioral abnormalities[29]. Recently, a long poly-GR mouse model, which was infected with adeno-associated virus (AAV) containing 100 repeats of GR, shows neurodegeneration and behavioral defects[30]. Expression of DPRs in primary cultured neurons and flies, especially poly-PR and poly-GR, is toxic to neurons[23]. However, it remains unclear whether the poly-PR is sufficient to induce neurodegeneration and behavioral changes in mice.

Here, we establish GFP-PR$_{28}$ transgenic mice which express poly-PR under the control of neuronal Thy1 promoter. Our results show that GFP-PR$_{28}$ homozygous mice decrease survival, while GFP-PR$_{28}$ heterozygous mice develop DPR inclusions and show atrophy of the cerebral cortex and loss of Purkinje cells in the cerebellum and motor neurons in the spinal cord. The GFP-PR$_{28}$ heterozygous mice have motor imbalance and ataxia-like phenotype. Gene Ontology (GO) analyses after RNA sequencing suggest a dysregulation of synaptic transmission-related genes and activation of inflammation in the GFP-PR$_{28}$ heterozygous mice.

## Results

**Distribution of GFP-PR$_{28}$ in the heterozygous mice**. To elucidate the pathological characters of poly-PR underlying c9FTD/ALS, we established GFP-PR$_{28}$$^{flox/flox}$ transgenic mice by inserting the GFP-PR$_{28}$ construct into the Rosa26 site. We bred GFP-PR$_{28}$$^{flox/flox}$ mice with Thy1-Cre mice, so that the stop codon before GFP-PR$_{28}$

was eliminated by Cre recombinase, leading to a specific expression of GFP-PR$_{28}$ in neurons under the control of Thy1 promoter (Fig. 1a, b). After crossing, we got the GFP-PR$_{28}$$^{flox/+}$; Thy1-Cre$^{+/−}$ mice and GFP-PR$_{28}$$^{flox/+}$; Thy1-Cre$^{−/−}$ mice. The GFP-PR$_{28}$$^{flox/+}$; Thy1-Cre$^{+/−}$ mice were the heterozygotes expressing poly-PR, and the littermates of GFP-PR$_{28}$$^{flox/+}$;Thy1-Cre$^{−/−}$ mice were used as control. The produced GFP-PR$_{28}$$^{flox/+}$; Thy1-Cre$^+$ mice (heterozygote) were further crossed with GFP-PR$_{28}$$^{flox/flox}$ transgenic mice to produce GFP-PR$_{28}$$^{flox/flox}$; Thy1-Cre$^+$ mice (homozygote).

We examined the distribution of GFP-PR$_{28}$ aggregates in different regions of GFP-PR$_{28}$ heterozygous mouse brain at 2 months of age. The intranuclear aggregates of GFP-PR$_{28}$ were presented in the hippocampus, motor cortex, brainstem, and cerebellum of GFP-PR$_{28}$ heterozygous mice (Fig. 1c), and the aggregates co-localize with nucleolin (Supplementary Fig. 1a, b). Moreover, the expression of GFP-PR$_{28}$ was also confirmed by immunohistochemical analysis with poly-PR antibody (Supplementary Fig. 1c). Besides intranuclear aggregates, a diffused cytoplasmic distribution of GFP-PR$_{28}$ was also observed in lumbar motor neurons (Fig. 1c; Supplementary Fig. 1d). Surprisingly, in the cerebellum, the expression of GFP-PR$_{28}$ was majorly observed in Purkinje cell layer, but little expression in the granular layer and molecular layer (Fig. 1c). To exclude the effects of Thy1 promoter on cellular distribution of GFP-PR$_{28}$ in the cerebellum, the distribution of Cre was examined using immunostaining with Cre recombinase antibody. Immunostaining showed an extensive expression of Cre recombinase in the cerebellum of 2-month-old Thy1-Cre transgenic mice, abundant in the granular layer and Purkinje cell layer (Supplementary Fig. 1e). The expression pattern in the cerebellum was further confirmed using immunostaining with antibody against calbindin, which is a specific marker of Purkinje cells in the cerebellum (Supplementary Fig. 1f). Moreover, the aggregates were presented in Neu-N-positive neurons (Supplementary Fig. 1g), but not in GFAP or Iba1-positive glia (Fig. 1d). Using quantitative real-time PCR, the GFP-PR$_{28}$ was identified to be expressed in the brain but not in peripheral tissues (Fig. 1e), further suggesting a neuronal-specific expression.

We also examined the distribution of GFP-PR$_{28}$ in the cerebellum and motor cortex of GFP-PR$_{28}$ heterozygous mice at 12 months of age. Poly-PR remained intranuclear and formed aggregates in both regions (Fig. 1f). The percentage of cells with GFP-PR$_{28}$ aggregates in major brain regions of 2-month-old transgenic mice were examined. Among the four regions, cerebellar Purkinje cells had the highest number harboring aggregates of poly-PR (Fig. 1g). Similar results were obtained in the heterozygous mice of GFP-PR$_{28}$ at 12 months of age (Fig. 1g).

**Homozygous mice display decreased survival time**. To determine the effects of GFP-PR$_{28}$ on mice, we examined the phenotypes of GFP-PR$_{28}$ homozygous mice. Surprisingly, the homozygous mice showed smaller body size accompanied by smaller brain volumes at 20 days of age than control mice (Fig. 2a; Supplementary Fig. 2a). GFP-PR$_{28}$ homozygous mice displayed lower body weight at 20 days of age than control mice (Fig. 2b), while no significant difference between control and GFP-PR$_{28}$ heterozygous mice was observed at that age, suggesting a possibility that the expressing GFP-PR$_{28}$ at high levels is highly toxic. Furthermore, the expressing GFP-PR$_{28}$ largely shortened the lifespan of the homozygous mice, since the homozygous mice died prematurely, averagely at day 36 after birth. In contrast, the control mice exhibited normal longevity (Fig. 2c).

**Heterozygous mice show motor deficits**. As GFP-PR$_{28}$ homozygous mice showed obvious decreased survival, the GFP-PR28$^{flox/+}$;

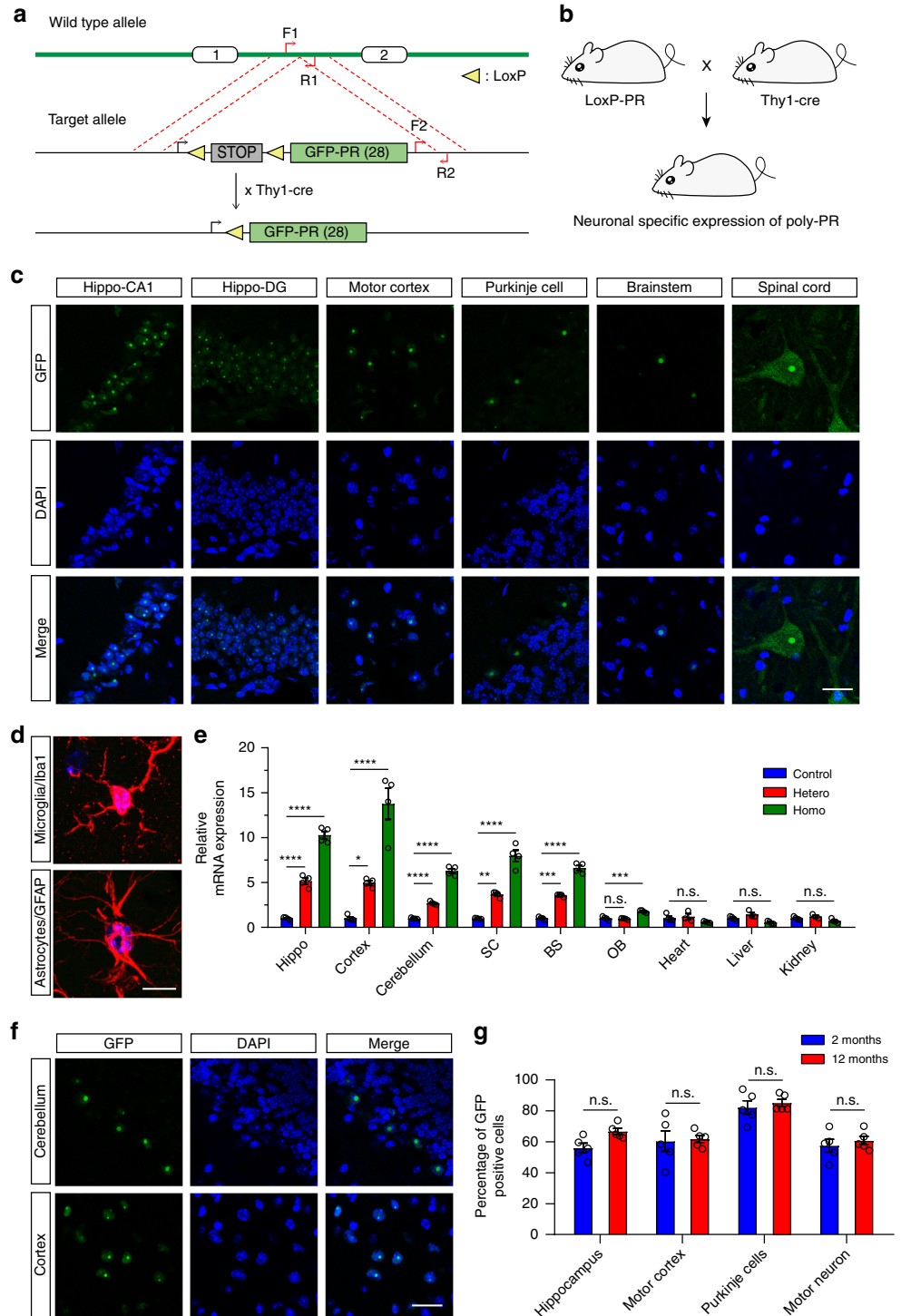

**Fig. 1** Distribution of GFP-PR$_{28}$ in heterozygous mice. **a** Diagram of the construct containing *GFP-PR$_{28}^{flox/flox}$* in *Rosa26* site. **b** Breeding scheme for producing *GFP-PR$_{28}^{flox/+}$;Thy1-Cre +* heterozygous mice driven by *Thy1* promoter for neuronal expressions. **c** Representative images showing distribution of poly-PR aggregates in different brain regions of GFP-PR$_{28}$ heterozygous mice at 2 months of age. GFP (green), Hoechst (blue). Scale bar represents 50 μm. **d** Representative images showing the distribution of GFP-PR$_{28}$ in GFAP or Iba1-positive glia in the motor cortex of 6-month-old GFP-PR$_{28}$ heterozygous mice. GFP (green), Hoechst (Blue), GFAP/Iba1 (Red). Scale bar represents 10 μm. **e** Relative mRNA levels of GFP in different brain regions and tissues of 20-day-old control, GFP-PR$_{28}$ heterozygous and homozygous mice. Hippocampus (Hippo), spinal cord (SC), brainstem (BS), olfactory bulb (OB). One-way ANOVA, Bonferroni post hoc test; *n* = 3–4 mice per group. **f** Representative images showing the distribution of poly-PR aggregates in the cerebellum and motor cortex of GFP-PR$_{28}$ heterozygous mice at 12 months of age. GFP (green), Hoechst (Blue). Scale bar represents 50 μm. **g** Percentage of cells containing GFP-PR$_{28}$ in different brain regions of GFP-PR$_{28}$ heterozygous mice at 2 and 12 months of age. Two-way ANOVA, Bonferroni post hoc test; *n* = 5 mice per group. All data were displayed as mean ± s.e.m. *$P$ < 0.05, **$P$ < 0.01, ***$P$ < 0.001, ****$P$ < 0.0001, n.s. not significant

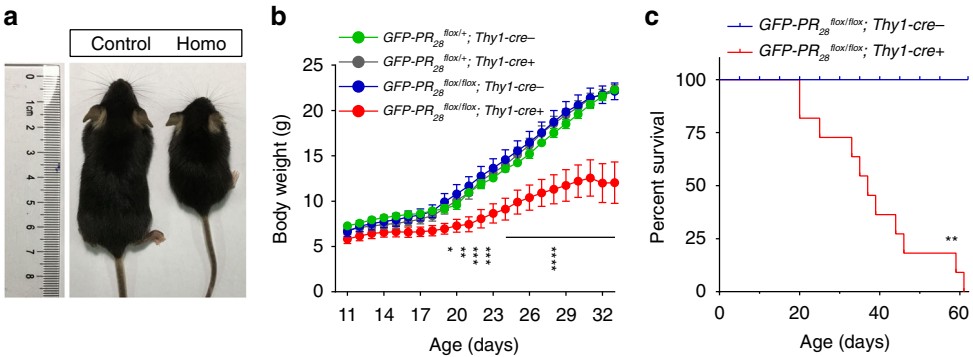

**Fig. 2** GFP-PR$_{28}$ homozygous mice display reduced body weight and decreased survival. **a** Representative image showing the body size of the control and homozygous mice at 20 days of age. **b** Body weight curves of control, heterozygous, and the homozygous mice from postnatal day 11 to 33. Two-way ANOVA, Bonferroni post hoc test; $n = 4, 6, 6, 6$ mice. **c** Kaplan–Meier survival curve of the control and homozygous mice. Gehan–Breslow–Wilcoxon test, $n = 12, 11$ mice. Control: GFP-PR$_{28}$$^{flox/flox}$;Thy1-Cre-, Homo: GFP-PR$_{28}$$^{flox/flox}$;Thy1-Cre + (**c**). All data are displayed as mean ± s.e.m. *$P < 0.05$, **$P < 0.01$, ***$P < 0.001$, ****$P < 0.0001$

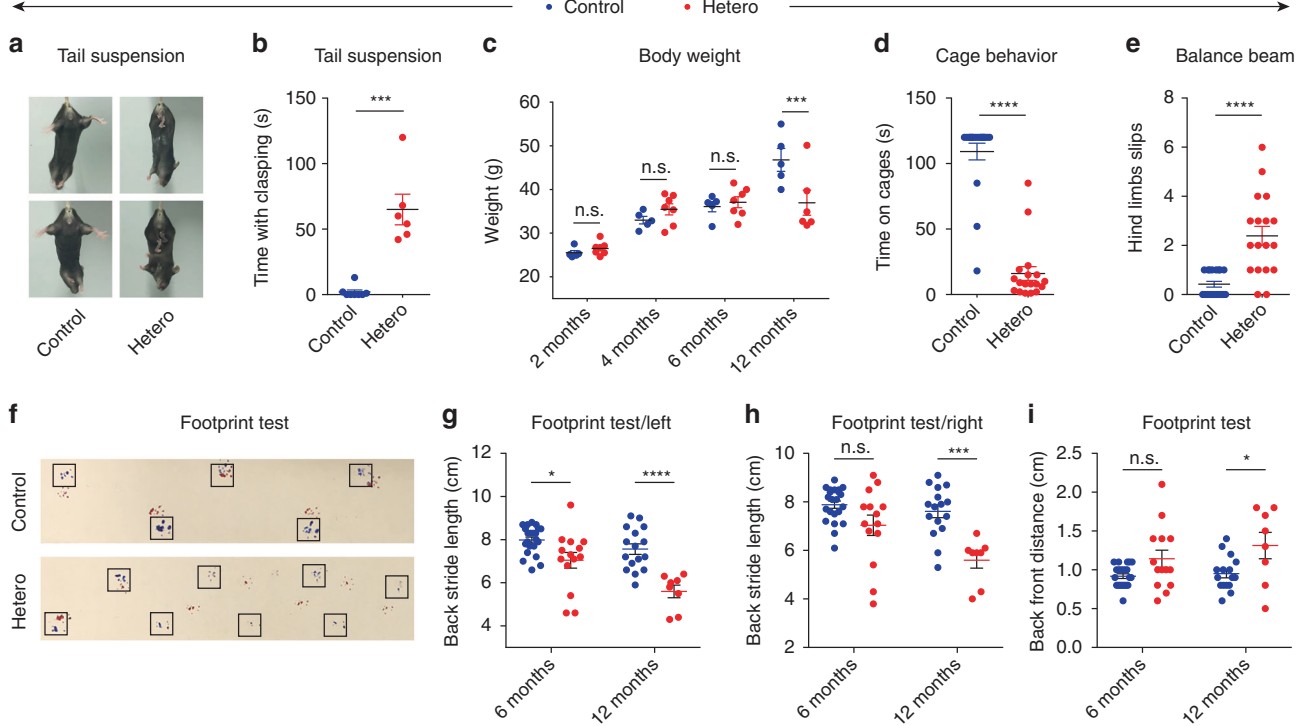

**Fig. 3** GFP-PR$_{28}$ heterozygous mice show motor deficits. **a** Representative images of male control and GFP-PR$_{28}$ heterozygous mice at 2 months of age in tail-suspension test. **b** Quantification of the clasping time of mice in (**a**) during 2-min test. Mann–Whitney test, $n = 8, 6$ mice. **c** Body weight of male control and GFP-PR$_{28}$ heterozygous mice at 2, 4, 6, and 12 months old. Two-way ANOVA, Bonferroni post hoc test; $n = 5, 7$ mice. **d** Quantification of time keeping on the edges of cages of male control and GFP-PR$_{28}$ heterozygous mice at 6 months of age. Mann–Whitney test, $n = 19, 18$ mice. **e** The numbers of hind limb foot slips on the balance beam test of male control and GFP-PR$_{28}$ heterozygous mice at 6 months of age. Mann–Whitney test, $n = 19, 18$ mice. **f** Representative images of male control and GFP-PR$_{28}$ heterozygous mice at 12 months of age in footprint test. Fore paws (red), hind paws (blue). Black squares indicate localization of hind paws. **g** Back stride length (left) of 6-, 12-month-old male control and GFP-PR$_{28}$ heterozygous mice measured in footprint test. Two-way ANOVA, Bonferroni post hoc test; 6 months, $n = 22, 14$ mice; 12 months, $n = 16, 8$ mice. **h** Back stride length (right) of 6-, 12-month-old male control and GFP-PR$_{28}$ heterozygous mice measured in footprint test. Two-way ANOVA, Bonferroni post hoc test; 6 months, $n = 22$, 14 mice; 12 months, $n = 16, 8$ mice. **i** Back-front distance measured in footprint test of male control and GFP-PR$_{28}$ heterozygous mice at 6, 12 months of age. Two-way ANOVA, Bonferroni post hoc test; 6 months, $n = 22, 14$ mice; 12 months, $n = 16, 8$ mice. All data are displayed as mean ± s.e.m. *$P < 0.05$, ***$P < 0.001$, ****$P < 0.0001$, n.s. not significant

*Thy1-Cre*$^+$ (heterozygote) mice were utilized for exploring the pathological and behavioral features. To determine whether the heterozygous mice expressing GFP-PR$_{28}$ show motor defects, we first performed tail-suspension test. Hind limbs clasping and sustained trembling were found in GFP-PR$_{28}$ heterozygous mice at 2 months of age, but not in control mice (Fig. 3a). We supposed that

motor neurons are involved in the phenotypes, so we performed hind and fore limbs grip strength test. Surprisingly, GFP-PR$_{28}$ heterozygous mice performed as well as control mice at 6 months of age (Supplementary Fig. 2b, c). Meanwhile, there was no significant difference in the body weight between control and GFP-PR$_{28}$ heterozygous male mice at 2, 4, and 6 months of age (Fig. 3c). However,

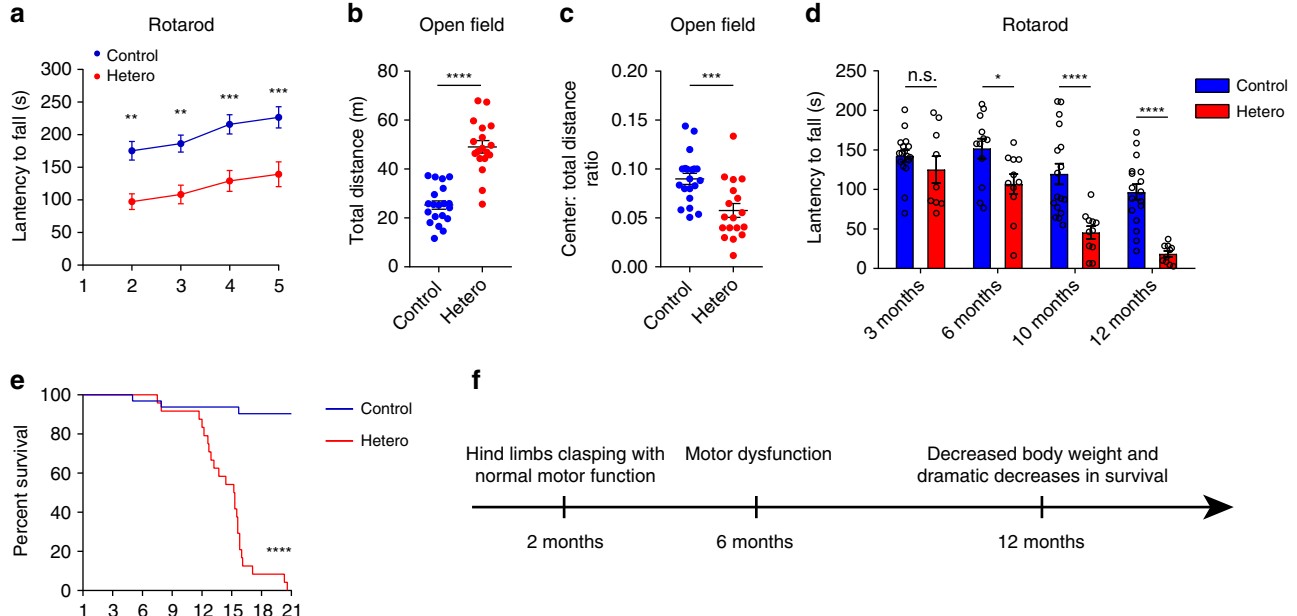

**Fig. 4** GFP-PR28 heterozygous mice show progressive motor deficits and anxiety-like behaviors. **a** Latencies to fall from the accelerated rotating beams of male control and GFP-PR28 heterozygous mice at 6 months of age. Two-way ANOVA, Bonferroni post hoc test; $n = 18$, 18 mice. **b**, **c** Open-field test showing the ratio of time in the center area and total ambulatory distance of male control and GFP-PR28 heterozygous mice at 6 months of age. Two-tailed $t$ test, $n = 20$, 18 mice. **d** Latencies to fall from the accelerated rotating beams of control and GFP-PR28 heterozygous mice at 3, 6, 10, and 12–16 months of age. Two-way ANOVA, Bonferroni post hoc test; 3 months, $n = 17$, 9 mice; 6 months, $n = 12$, 11 mice; 10 months, $n = 17$, 11 mice; 12–16 months, $n = 17$, 10 mice. **e** Kaplan–Meier survival curve of control and heterozygous mice. Gehan–Breslow–Wilcoxon test, $n = 32$, 24 mice. **f** Diagram of the disease progression of GFP-PR28 heterozygous mice. All data are displayed as mean ± s.e.m. *$P < 0.05$, ***$P < 0.001$, ****$P < 0.0001$, n.s. not significant

at 12 months of age, the body weight of male heterozygous mice significantly decreased as compared with control mice (Fig. 3c).

Using a cage behavior test, a simple and elegant way to assay motor coordination, we found that GFP-PR28 heterozygous mice, at 6 months of age, fell down within 2 min when walked along the edges of cage, but control mice did not (Fig. 3d; Supplementary Movies 1, 2), suggesting that GFP-PR28 heterozygous mice developed motor imbalance. Therefore, we conducted rotarod test, balance beam test, and footprint test, which were three widely used tests to measure motor functions. GFP-PR28 heterozygous mice showed significantly increased hind limb slips on the balance beam (Fig. 3e), age-dependent gait abnormalities (Fig. 3f–i), and decreased latency to fall off the rotarod (Fig. 4a). Similar results were obtained using female heterozygous mice (Supplementary Fig. 2d–h).

We also performed open-field test to measure locomotion activity. GFP-PR28 heterozygous mice displayed a longer distance exploring the novel environment, with decreased time spent in the center region of the chamber compared with control mice (Fig. 4b, c), indicating hyperactivity and anxiety-like behaviors. To determine the disease progression of GFP-PR28 heterozygous mice, motor performance was evaluated at 3, 6, 10, and 12–16 months of age. Motor deficiency was observed at 6 months of age and deteriorated at 10 months of age (Fig. 4d). In addition, the GFP-PR28 heterozygous mice showed dramatically decreased survival between 12 and 18 months of age (Fig. 4e). The disease progression is summarized in Fig. 4f.

**Heterozygous mice show motor-related neurodegeneration.** Given that GFP-PR28 heterozygous mice developed motor behavior defects, we examined brain weight and cerebellum weight, as well as the body weight of mice at 2, 5, and 12 months of age. Despite no significant difference of body weight between control and GFP-PR28 heterozygous mice until 12 months old

(Figs 2b, 3c; Supplementary Fig. 3a), the brain weight or cerebellum weight was significantly reduced as early as at age of 2 months (Supplementary Fig. 3b, c). Immunohistochemical staining further confirmed the atrophy of the cerebellum and decreased thickness of the molecular layer (Fig. 5a, b). As the expression of GFP-PR28 was specifically localized in Purkinje cells of the cerebellum, and GFP-PR28 transgenic mice showed obvious motor imbalance, we evaluated the numbers of Purkinje cell in the cerebellum. Interestingly, immunostaining showed a significantly decreased numbers of Purkinje cell in 6-month-old GFP-PR28 heterozygous mice compared with control mice (Fig. 5c, d).

As motor behavior defects can be attributed to the loss of upper and lower motor neurons, we evaluated the number of motor neurons in the motor cortex and lumbar spinal cord at 2, 6, and 12 months of age. Unexpectedly, the thickness of the motor cortex significantly decreased in GFP-PR28 heterozygous mice (Fig. 5e, f). Moreover, immunohistochemical staining showed loss of ChAT-positive lower motor neurons in the spinal cord of GFP-PR28 heterozygous mice (Fig. 5g, h). In addition, we also calculated the number of hippocampal neurons in 6-month-old control and GFP-PR28 heterozygous mice using Hoechst staining, no hippocampal neuronal loss was observed (Supplementary Fig. 4a, b). Given that TDP-43 cytoplasmic inclusions are major neuropathological feature of c9FTD/ALS[31,32], we examined the cellular distribution of TDP-43 in the cerebellum and lumbar spinal cord of control and heterozygous mice at 12 months of age. Immunohistochemical staining showed no cellular inclusion of TDP-43 in the Purkinje cells of the cerebellum and lower motor neurons (Supplementary Fig. 4c).

**Gliosis in heterozygous mice.** Gliosis is a major neuropathological feature of neurodegenerative diseases[33], we wonder whether glia are activated in GFP-PR28 heterozygous mice.

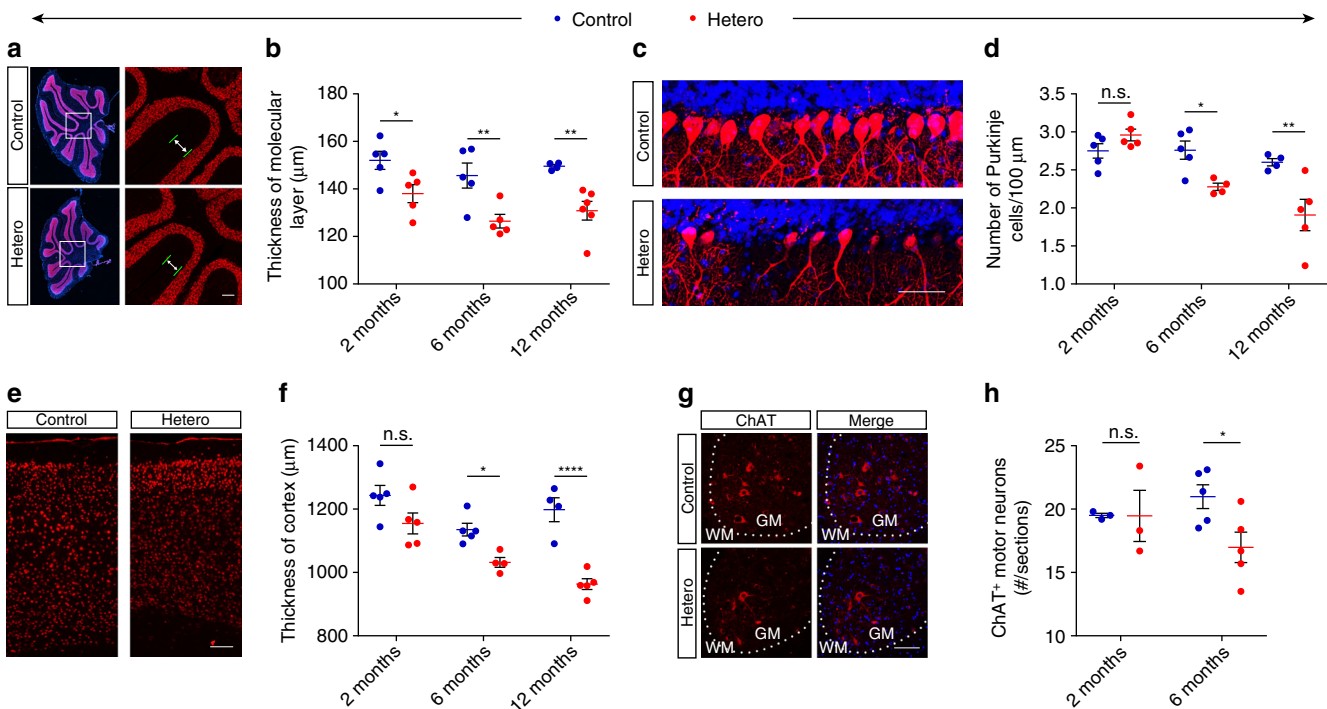

**Fig. 5** Expression of GFP-PR$_{28}$ causes motor-related neurodegeneration. **a** Representative images showing the size of the cerebellum and the thickness of the molecular layer of control and GFP-PR$_{28}$ heterozygous mice at 6 months of age. Neu-N (red), Hoechst (blue). White squares indicates enlarged area. Green lines indicate the thickness of the molecular layer. Scale bar represents 100 μm. **b** Quantification of the thickness of molecular layer of control and GFP-PR$_{28}$ heterozygous mice at 2, 6, and 12 months of age. Two months, $n = 5$, 5 mice; 6 months, $n = 5$, 5 mice; 12 months, $n = 4$, 6 mice. **c** Representative images showing the numbers of cerebellar Purkinje cells of 6-month-old control and GFP-PR$_{28}$ heterozygous mice. Calbindin (red), Hoechst (blue). Scale bar represents 50 μm. **d** Quantification of the numbers of calbindin positive Purkinje cells of control and GFP-PR$_{28}$ heterozygous mice at 2, 6 and 12 months of age. Two months, $n = 5$, 5 mice; 6 months, $n = 5$, 5 mice; 12 months, $n = 4$, 5 mice. **e** Representative images showing the thickness of motor cortex of control and GFP-PR$_{28}$ heterozygous mice at 6 months of age. Scale bar represents 100 μm. **f** Quantification of the thickness of the motor cortex of control and GFP-PR$_{28}$ heterozygous mice at 2, 6, and 12 months of age. Two months, $n = 5$, 5 mice; 6 months, $n = 5$, 4 mice; 12 months, $n = 4$, 5 mice. **g** Representative images showing the numbers of ChAT-positive motor neurons in the lumbar spinal cord of 6-month-old control and GFP-PR$_{28}$ heterozygous mice. GM (gray matter), WM (white matter). Scale bar represents 100 μm. **h** Quantification of the numbers of ChAT-positive motor neurons of control and GFP-PR$_{28}$ heterozygous mice at 2 and 6 months of age. Two months, $n = 3$, 3 mice; 6 months, $n = 5$, 5 mice. All data are displayed as mean ± s.e.m. Two-way ANOVA, Bonferroni post hoc test; *$P < 0.05$, **$P < 0.01$, ***$P < 0.001$, ****$P < 0.0001$, n.s. not significant

Immunohistochemical staining showed higher integrated optical densities of GFAP (astrocyte marker) (Fig. 6a) and Iba1 (microglial marker) (Fig. 6b) in the spinal cord of 6-month-old GFP-PR$_{28}$ heterozygous mice compared with control mice (Fig. 6a, b). Unexpectedly, activation of astrocytes in the spinal cord was presented at 2 months of age (Fig. 6c, d), although no obvious neuronal deficiency was observed, suggesting a possibility that neurodegeneration may occur as early as at 2 months of age. The activation of astrocytes was also observed in the cerebellum of GFP-PR$_{28}$ heterozygous mice at 6 months of age (Fig. 6e). Quantitative data for the signal densities of cerebellar astrocytes showed age-dependent activation of inflammation (Fig. 6f). Increased expression of *Gfap* and *Iba1* mRNA further verified the activation of astrocytes and microglia in the cerebellum (Supplementary Fig. 4d, e). However, only a few of GFAP-positive astrocytes in the motor cortex were observed (Supplementary Fig. 4f), and no gliosis in the hippocampus of GFP-PR$_{28}$ heterozygous mice (Supplementary Fig. 4g).

**RNA-seq analyses of dysregulated genes in heterozygous mice.** To explore the pathological mechanisms that GFP-PR$_{28}$ causes neuronal deficiency, we isolated RNA from the cerebellum of 5-month-old mice and performed RNA sequencing. We identified 239 genes that were upregulated and 399 genes that were

downregulated in GFP-PR$_{28}$ transgenic mice relative to control mice (Fig. 7a, b; Supplementary Data 1). GO analyses of enriched categories among upregulated genes were translation, immune system process, and innate immune process, which further confirmed the activation of microglia and astrocytes (Fig. 6e, f). Enriched categories among downregulated genes were calcium ion-regulated exocytosis of neurotransmitter, intracellular signal transduction, and neurotransmitter secretion (Fig. 7c). Moreover, the Reactome pathway analyses identified transmission across chemical synapses that was a major pathway implicated in cerebellar pathology (Fig. 7d). We validated the downregulation of candidate genes (*Camk4, Grin2a, Kcnj9, Rims3, Syt2,* and *Unc13a*) associated with synaptic transmission using qRT-PCR (Fig. 7e). Western blotting assay further confirmed the downregulation of CaMK IV labeled with antibody against CaMK IV (Fig. 7f). These data indicate ongoing neurodegeneration in cerebellar neurons of GFP-PR$_{28}$ transgenic mice.

To exclude the influence of Purkinje cell loss on downregulation of genes associated with synaptic transmission, we performed RNA sequencing using RNA from 2-month-old mouse cerebellum, in which no degeneration of Purkinje cell was observed with immunohistochemical staining (Supplementary Fig. 5a, Supplementary Data 2). We identified 11 genes that were upregulated and 31 genes that were downregulated (Supplementary Fig. 5b, c). The genes dysregulated in the

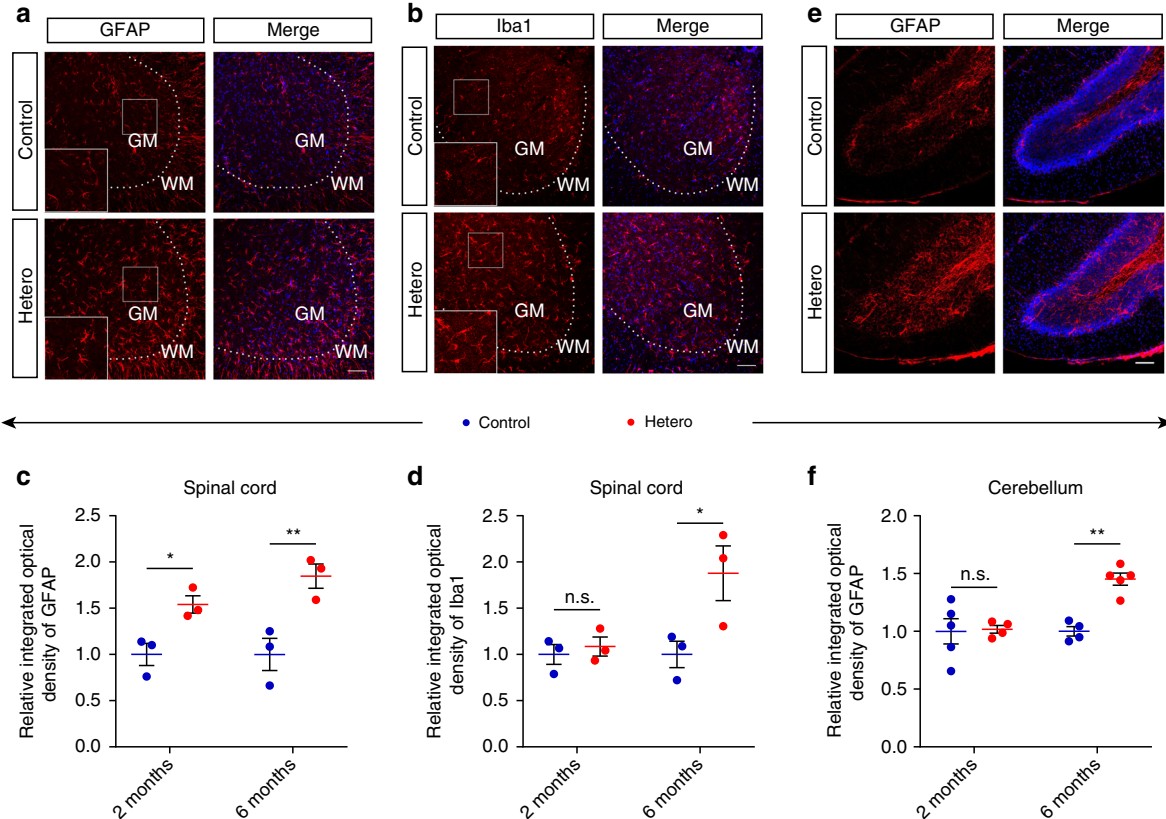

**Fig. 6** Gliosis in the lumbar spinal cord and cerebellum of GFP-PR$_{28}$ heterozygous mice. **a** Representative images showing the numbers of GFAP-positive astrocytes in the lumbar spinal cord of 6-month-old control and GFP-PR$_{28}$ heterozygous mice. GM (gray matter), WM (white matter). White squares are enlarged views of corresponding images. GFAP (red), Hoechst (blue). Scale bar represents 100 μm. **b** Representative images showing the numbers of Iba1-positive microglia in the lumbar spinal cord of 6-month-old control and GFP-PR$_{28}$ heterozygous mice. GM (gray matter), WM (white matter). White squares are enlarged views of corresponding images. Iba1 (red), Hoechst (blue). Scale bar represents 100 μm. **c** Relative integrated optical density of GFAP in the lumbar spinal cord of control and heterozygous mice at 2 and 6 months of age. Two-way ANOVA, Bonferroni post hoc test; 2 months, $n = 3$, 3 mice; 6 months, $n = 3$, 3 mice. **d** Relative integrated optical density of Iba1 in the lumbar spinal cord of control and heterozygous mice at 2 and 6 months of age. Two-way ANOVA, Bonferroni post hoc test; 2 months, $n = 3$, 3 mice; 6 months, $n = 3$, 3 mice. **e** Representative images showing the numbers of GFAP-positive astrocytes in the cerebellum of 6-month-old control and GFP-PR$_{28}$ heterozygous mice. GFAP (red), Hoechst (blue). Scale bar represents 100 μm. **f** Relative integrated optical density of GFAP in the cerebellum of control and heterozygous mice at 2 and 6 months of age. Two-way ANOVA, Bonferroni post hoc test; 2 months, $n = 5$, 4 mice; 6 months, $n = 4$, 5 mice. All data are displayed as mean ± s.e.m. *$P < 0.05$, **$P < 0.01$, n.s. not significant

cerebellum of 2-month-old mice were highly associated with those in 5-month-old mice (Supplementary Fig. 5d). Moreover, calcium ion-regulated exocytosis of neurotransmitter was also a major pathway among downregulated genes identified by GO analyses (Supplementary Fig. 5e). Two downregulated genes (*Rims3*, *Doc2b*) related to synaptic function were identified using RNA-sequencing analysis and validated with qRT-PCR (Supplementary Fig. 5f, g).

Given that the motor cortex and spinal cord were major brain regions degenerated in patients, we also performed RNA sequencing using RNA from 6-month-old mice cortex and lumbar spinal cord (Supplementary Fig. 6a–f, Supplementary Data 3, 4). GO analyses of enriched categories identified positive regulation of neurotransmitter secretion and exocytosis that were the major pathways implicated in cortical pathology (Supplementary Fig. 6c), this is highly consistent with the results of synaptic-related genes dysregulation in the cerebellum of heterozygous mice (Fig. 7c–e). GO analyses of enriched categories identified immune system process and innate immune process were major pathway implicated in the spinal cord (Supplementary Fig. 6f), and a significant overlap of differentially expressed genes between the cerebellum and spinal cord was identified (Supplementary Fig. 6g, h). Three upregulated genes (*C1qa*, *Cd68*,

and *Trem2*) related to inflammation were further validated with qRT-PCR (Supplementary Fig. 6i), and these results are consistent with the increased activation of glia in the cerebellum and spinal cord (Fig. 6a–f).

## Discussion

We generated a mouse model that the transgenic animals specifically expressed poly-PR (GFP-PR$_{28}$), but without repeat RNA in neurons driven by *Thy1* promoter. The GFP-PR$_{28}$ homozygous mice showed reduced body size, decreased body weight, and reduced premature survival. GFP-PR$_{28}$ heterozygous mice showed motor deficits, especially in progressive gait and balance impairment. Consistent with abnormal behaviors that are associated with the cerebellum, loss of Purkinje cells, but not hippocampal neurons, were presented in GFP-PR$_{28}$ heterozygous mice. Moreover, microglia and astrocytes in the cerebellum and lumbar spinal cord of GFP-PR$_{28}$ heterozygous mice were significantly activated. Finally, the poly-PR expressing neurons developed synaptic transmission-related genes dysregulation.

Although c9FTD/ALS pathogenesis may be associated with multiple mechanisms, including loss of function or gain toxic functions by formation of RNA foci or expression of DPRs, the

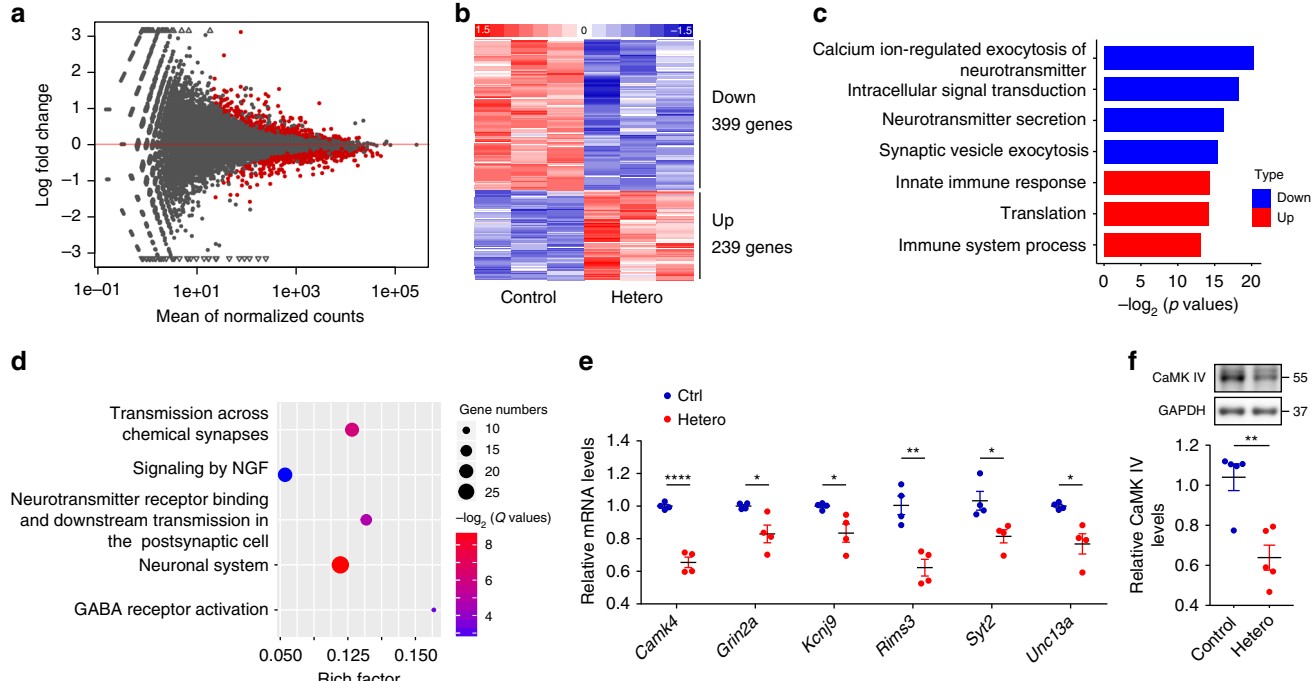

**Fig. 7** GFP-PR$_{28}$ expression is associated with synaptic transmission-related genes dysregulation. **a** MA-plot of differentially expressed genes in the cerebellum of 5-month-old control and GFP-PR$_{28}$ heterozygous mice. Red blots indicate significant changes, $n = 3$, 3 mice. **b** Hierarchical clustering of differentially expressed genes in the cerebellum of 5-month-old control and GFP-PR$_{28}$ heterozygous mice. **c** Gene ontology (GO) biological processes analyses of upregulated and downregulated genes in (**b**). **d** The Reactome pathway analyses of top five enriched terms of downregulated genes in (**b**). Gene numbers indicate genes that are enriched in this pathway. Rich factor indicates the ratio of enriched genes to total genes in this pathway. **e** Relative mRNA expression of six genes (*Camk4, Grin2a, Kcnj9, Rims3, Syt2*, and *Unc13a*) in association with synaptic transmission in the cerebellum of 5-month-old control and GFP-PR$_{28}$ heterozygous mice. Two-tailed *t* test, $n = 4$, 4 mice. **f** Relative expressing levels of CaMK IV in the cerebellum of 6-month-old control and GFP-PR$_{28}$ heterozygous mice, determined by western blotting assay. Two-tailed *t* test, CaMK IV, $n = 5$, 5 mice. All data are displayed as mean ± s.e.m. *$P < 0.05$, **$P < 0.01$, ****$P < 0.0001$

expression of DPRs is known as one of the causative factor for c9FTD/ALS, evidenced by identification of DPR pathological features in patient brains[17,34–36], and expression of DPR without hexanucleotide-repeat RNA in animal models[27,29]. However, it is still unknown whether DPR species differentially induce animal phenotype correlate to regionally neuronal loss in animal models. Previous studies have indicated that arginine-rich poly-PR is highly toxic to cells in vitro[23,24] and in transgenic Drosophila models[23,25,27,28]. Using our poly-PR mouse model, we are able to explore the role of poly-PR in vivo and to identify susceptible neurons to poly-PR toxicity.

Surprisingly, GFP-PR$_{28}$ homozygous mice showed significantly premature death, which is not observed in BAC transgenic mice of c9FTD/ALS[12,18–20]. However, this phenotype made us reminisce of the premature death of PR$_{50}$ expressed flies[23], further suggesting a high toxicity of poly-PR in vivo. While, our GFP-PR$_{28}$ heterozygous mice showed no obviously shortened lifespan until 12 months old, indicating that poly-PR causes neuronal toxicity in a dose-dependent manner.

Due to the largely low survival of GFP-PR$_{28}$ homozygous mice, we evaluated the motor function using GFP-PR$_{28}$ heterozygous mice. GFP-PR$_{28}$ heterozygous mice showed obvious motor imbalance and anxiety-like behavior, which is highly consistent with previous studies[18,29,37] and clinical features of c9FTD/ALS[38,39]. Moreover, the GFP-PR$_{28}$ heterozygous mice developed deficiency of motor performance at 6 months of age, while no obvious deficits at 3 months of age, suggesting that poly-PR causes neuronal deficiency in an age-dependent manner. Despite body weight remained no changes in 2-month-old GFP-PR$_{28}$ heterozygous mice compared with control mice, the brain weight

was largely reduced in GFP-PR$_{28}$ heterozygous mice. Notably, the decreased ratio of cerebellar weight (the heterozygous mice vs. control) was higher than that of whole brain, indicating that the cerebellar deficiency may be a key contributor to motor coordination defects. Consistently, pure cerebellar ataxia has been found in the patients with *C9ORF72* hexanucleotide-repeat expansion mutation[39–41], indicating a direct relationship between *C9ORF72* mutation and cerebellar defects. Furthermore, GFP-PR$_{28}$ heterozygous mice presented a significant Purkinje cells loss, while the number of hippocampal neurons kept unchanged.

In a poly-GA transgenic mouse model, in which a specifically neuronal expression is driven by *Thy1* promoter, the animals developed gait and balance impairment with inflammation in the lumbar spinal cord[42]. Interestingly, the phenotypes and pathological changes observed in our mouse model are very similar to a poly-PR mouse model with GFP-PR$_{50}$ AAV1 infected[43]. The typically behavioral changes of animals in both models are motor dysfunction, with reduced brain weight and decreased numbers of Purkinje cell. Moreover, a poly-GR AAV-infected animal model also shows defects in the cerebellum, motor cortex and hippocampus[30]. Taken together with data from mouse models of (GGGGCC)$_{66}$[37], (GR)$_{100}$[30], and (PR)$_{50}$[43], which also show loss of Purkinje cells, we propose that the DPR species, at least arginine-rich poly-GR and poly-PR, may have similar susceptible subtypes of neurons, leading to similar phenotypes.

Given that poly-PR transgenic mice showed obvious neuropathology, we performed RNA sequencing to identify pathological mechanisms. Synaptic transmission-related genes were significantly downregulated in the cerebellum of heterozygous mice, suggesting an ongoing neurodegeneration. Unfolded

protein response (UPR) was identified as a major module both in the cerebellum and frontal cortex in c9ALS[44]. In addition, endoplasmic reticulum (ER) stress was also found in poly-GA-infected primary cortical neurons[45]. Although ER-stress was not enriched in our poly-PR transgenic mice using GO pathway analyses, the major genes Chac1 and Atf5 that reflect ER-stress were both upregulated in the cerebellum of 2-month- and 5-month-old heterozygous mice (Supplementary Fig. 5c, Supplementary Data 1), suggesting that ER-stress is an early event in poly-PR expressing neurons, which is highly consistent with the data from others[46]. Poly-PR-transfected primary cortical neurons showed upregulation of ER-stress-related genes[46]. Thus, ER-stress may be a common pathological mechanism in poly-PR expressing neurons.

In spite of significant intranuclear aggregates of poly-PR in neurons, no cytoplasmic TDP-43 inclusions were identified in our GFP-PR$_{28}$ transgenic mice. Cytoplasmic TDP-43 inclusions were also absent in GFP-GA$_{50}$, GA$_{149}$-CFP, GFP-GR$_{100}$, and GFP-PR$_{50}$ transgenic mice[29,30,42,43], while the inclusions can be found in burden neurons of C9ORF72 BAC transgenic mice[18]. In addition, a correlation between antisense RNA foci and TDP-43 pathology in motor neurons of C9ORF72 patients was identified[47], suggesting that RNA foci may be a cause of TDP-43 inclusions. Given the lowrepeat length of our poly-PR construction, continuously expressed poly-PR with longer repeat length was needed. Another possibility is that combined RNA foci and DPRs may contribute to the pathology of TDP-43.

In conclusion, we have successfully generated poly-PR transgenic mice, presenting behavioral and neuropathological features of c9FTD/ALS. The mice demonstrated significant phenotype and pathological changes in association with the cerebellum, cortex, and spinal cord, which can be used as a tool for further investigating the role of poly-PR, the similarity and discrepancy of poly-PR with other DPR species.

## Methods

**Animals**. The GFP-PR$_{28}$^{flox/flox} transgenic mice were constructed by Beijing Bio-cytogen Co., Ltd. Briefly, the floxed GFP-PR$_{28}$ allele was generated by injecting the Cas9/sgRNA-embedded GFP-PR$_{28}$ vector into blastocysts derived from a C57BL/6 strain. The chimeras were first bred to a wild-type C57BL/6 mouse strain to generate mice with stabilized expression of GFP-PR$_{28}$. The GFP-PR$_{28}$ homozygous or heterozygous floxed mice were crossed with Thy1-Cre mice (Jackson Laboratory) for neuron-specific GFP-PR$_{28}$ overexpression. The primers in Supplementary Table 1 were used for genotyping. All animal experiments were approved by the Soochow University Institutional Animal Care and Use Committee, and were conducted in compliance with all relevant ethical regulations for animal testing and research.

**Construction of plasmids**. To construct plasmids GFP-PR$_{28}$, the CCAAGA oligonucleotide with 28 repeats was synthesized by Invitrogen and subcloned into the pEGFP-C1 vector (Clontech Laboratories) through EcoRI and BamHI sites. The fidelity of sequence of the plasmids was verified using sequencing (GENEWIZ).

**Western blotting analysis**. Isolated tissues were homogenized by PRO200 homogenizer (PRO Scientific) and lysed in 1 × cell lysis buffer (50 mM Tris-HCl (pH 7.5), 150 mM NaCl, 1% nonidet P40, and 0.5% sodium deoxycholate) supplemented with a protease inhibitor cocktail (Roche). Approximately 10–20 μg of protein per sample was separated on 10–12% SDS polyacrylamide gel, and then transferred onto a polyvinylidene difluoride (PVDF) membrane followed by blocking in 5% skimmed milk. The following primary antibodies were used: mouse monoclonal anti-CaMK IV antibody (sc-55501, 1:1000, Santa Cruz), mouse monoclonal anti-GAPDH antibody (MAB374, 1:2000, Millipore). The membranes were washed with TBST and incubated with following secondary antibodies: horseradish peroxidase (HRP)-conjugated goat anti-rabbit IgG (111-035-045, 1:10,000, Jackson ImmunoResearch Laboratories); HRP-conjugated goat anti-mouse IgG (115-035-062, 1:10,000, Jackson ImmunoResearch Laboratories). Peroxidase activity was detected with enhanced chemiluminescence substrate (ThermoFisher Scientific) and visualized with ChemiQ4800 imaging system (Bioshine). Uncropped blots are provided as a Source Data file.

**Immunohistochemical analysis**. Twenty micrometer-thick sagittal brain sections were prepared from mice at 2 and 6 months of age. Sections were incubated in blocking solution (1 × PBS with 0.4% Triton X-100 and 5% normal fetal bovine serum) for 1 h. The sections were then incubated with following primary antibodies in blocking buffer: mouse monoclonal anti-Neu-N antibody (MAB377, 1:500, Millipore), goat polyclonal anti-choline acetyltransferase (ChAT) antibody (AB144P, 1:300, Millipore), rabbit polyclonal anti-Iba1 antibody (019-19741, 1:1000, Wako Chemicals), mouse monoclonal anti-GFAP antibody (MAB360, 1:2000, Millipore), rabbit polyclonal anti-calbindin antibody (13176, 1:1000, Cell Signaling Technology), rabbit polyclonal anti-PR antibody (23979-1-AP, 1:300, proteintech), rabbit polyclonal anti-Cre recombinase antibody (15036, 1:500, Cell Signaling Technology), rabbit polyclonal anti-TDP-43 antibody (10782-2-AP, 1:1000, Proteintech), rabbit polyclonal anti-nucleolin antibody (ab129200, 1:1000, Abcam), mouse monoclonal anti-GFP antibody (sc-9996, 1:300, Santa Cruz). After washing, the sections were incubated with anti-mouse/rabbit secondary antibodies conjugated with either Alexa Fluor 488 (A21202, 1:300, ThermoFisher Scientific) or Alexa Fluor 594 (111-585-003; 1:300; Jackson ImmunoResearch Laboratories). Next, the brain sections were incubated with Hoechst 33342 dye (B2261, 1:1000, Sigma Aldrich) for 10 min. The autofluorescence was quenched by 0.1% Sudan black B (SBB) in 70% ethanol for 10 min at room temperature[48], followed by washing in PBS for 5 min. Finally, the sections were mounted with antifade reagent (Beyotime Biotechnology). The sections were imaged using either cooled CCD (DP72, Olympus) on a microscope (IX71, Olympus) or a laser confocal microscope (LSM 710, Carl Zeiss) with 63 × /1.40 oil DIC M27 objective. The images were captured and processed using software CellSens standard (Olympus) and Zen (Carl Zeiss).

**Quantification**. The percentage of cells with poly-PR aggregates in major brain regions was quantified manually. Briefly, 20-μm-thick sagittal brain sections were prepared from heterozygous mice, the percentage of poly-PR aggregates in the motor cortex was counted by percentage of GFP-positive cells in Neu-N-positive neuron. Similarly, the percentage of GFP-positive cells in ChAT-postive motor neurons and calbindin-positive Purkinje cellls were recorded, respectively. The percentage of GFP-positive cells in DAPI-positive hippocampal neurons (CA1–CA3) was also recorded. Five mice were determined for each brain regions, three sections were counted per mouse.

To quantify the numbers of Pukinje cell in the cerebellum, 40-μm-thick sagittal brain sections (just along to the midline) were prepared. The numbers of calbindin-positive Purkinje cell in the four and five lobes of the cerebellum were recorded using NIS elements (Nikon), and quantitative data were presented as the number of Purkinje cells per 100 μm, three slides per animal and 4–5 animals per group. The thickness of molecular layer in the four and five lobes of the cerebellum was measured in a similar way. Twenty micrometer-thick sagittal brain sections between lateral 0.72 mm and 1.08 mm were prepared for the quantification of the thickness of the motor cortex, matching sections through the motor cortex from bregma −1.0 mm to 0 mm were selected with three sections quantified for 4–5 mice, respectively. The hippocampal neurons in the squares indicated by Hoechst staining were analyzed using CellSense standard (Olympus). The number of hippocampal neurons in CA1, CA3, and DG regions was counted using Image J (National Institute of Health). The number of ChAT-positive motor neurons in lumbar spinal cord was counted manually. Briefly, 20-μm-thick lumbar sections (L2-L5) were prepared. The average numbers of motor neurons in nine sections were counted per mouse, 4–5 mice per group. The relative integrated optical densities of Iba1 and GFAP in the lumbar spinal cord were calculated in a similar manner. Briefly, images were converted to grayscale, and background was subtracted using default settings (50 pixels), consistent regions of interest in the ventral horn were selected using tool ROI manager (Image J, National Institute of Health). The relative integrated optical densities of Iba1 and GFAP in nine sections were counted per mouse, three mice per group. The relative integrated optical densities of GFAP in other brain regions were measured similarly (three sections per mouse, 4–5 mice per group).

**RNA sequencing and bioinformatics analyses**. The total RNA of each sample was extracted using TRIzol Reagent (Invitrogen). The total RNA with RNA integrity number (RIN) value above 9 was used for library preparation. The library for next-generation sequencing was constructed according to the manufacturer's protocol (NEBNext®Ultra™ RNA Library Prep Kit for Illumina®). The libraries with different indices were multiplexed and loaded on an Illumina HiSeq instrument according to the manufacturer's instructions (Illumina, San Diego, CA, USA). Sequencing was carried out using a 2 × 150 -bp paired-end (PE) configuration. Image analyses and base calling were conducted by the HiSeq Control Software (HCS) + OLB + GAPipeline-1.6 (Illumina) on the HiSeq instrument.

FASTQ files were trimmed with Trimmomatic (v0.30) to remove quality of bases lower than 20, then clean data were aligned to reference genome (Ensembl, mm10) via software Hisat2 (v2.0.1). The FASTA format was converted from known GFF annotation file and indexed properly. HTSeq (v0.6.1) was used to estimate gene and isoform expression levels from the pair-end clean data. Differentially expressed genes (DEGs) were selected using DESeq2[49], DEGs with adjusted P-values < 0.05 and |log2 (fold change)| > 0.2 were selected for further analyses. Hierarchical clustering of differentially expressed genes was analyzed

using pheatmap package in R studio (R Studio, Inc). GO pathway enrichment analyses were performed with DAVID v6.8[50,51], the Reactome enrichment pathway analyses were performed using Gene Ontology Consortium[52]. Top five enrichment pathways with adjusted $P$-values < 0.05 were selected.

**Quantitative real-time PCR (qRT-PCR).** Isolated tissues were homogenized in TRIzol reagent (Invitrogen) using PRO200 homogenizer (PRO Scientific). The total RNA was extracted according to the manufacturer's protocol. Five hundred nanograms of the total RNA were reverse-transcribed into cDNA with a Trans-Script First-Strand cDNA Synthesis Kit (Takara). Real-time PCR was performed with Power SYBR® Green PCR master mix (Applied Biosystems) and Supplementary Table 2 primers (most primers were searched in PrimerBank, GFP primers[29]). qRT-PCR was run on an ABI 7500 apparatus (Applied Biosystems). The expression of mRNA was measured via the $\Delta\Delta C_t$ method relative to that of an endogenous control gene ($\beta$-Actin). In addition, the expressions of synatic transmission-related genes were normalized to $Pcp2$ gene, which is specifically expressed in Purkinje cells of the cerebellum.

**Behavioral assays.** Two-, 6- and 12-month-old Thy1-Cre-positive heterozygous mice and littermate control mice were subjected to serials of behavioral tests. All mice were acclimated to the testing room for at least 1 h before formal test. No more than two tests each day were performed. The testing room was kept quietly and forbidden to disturb the spontaneous activity of mice.

**Tail-suspension test.** Two-month-old Thy1-Cre-positive heterozygous mice and control mice were suspended 50 cm above the surface of a table. Duration time with hind limb clasping was recorded in 2 min.

**Grip strength test.** The muscular strength of 20-day-old GFP-PR$_{28}$ homozygous mice, 6-month-old GFP-PR$_{28}$ heterozygous mice, and control mice were measured with a grip strength apparatus (Bioseb). After the mice grasped the grid with its hind paws or four paws, pulled backward gently until the mice released, the strength measured in gram was recorded, and the average strength of each mice was calculated in five intermittent measures.

**Cage behavior test.** Cage behavior test is a simple and elegant experiment to measure motor balance and coordination. Briefly, the mice were put on the edge of a cage, where is 15 cm above the surface of a table. Importantly, the mice should never be subjected to this test before. The performance of mice was recorded. The wild-type mice would walk along or stay on the edge constantly, while the ataxic mice would fall down due to motor imbalance. Duration time on the cage edge was recorded in 2 min.

**Rotarod test.** The mice were placed on the static beam (Ugo Basile), the latency fell from the beam was recorded when the beam was accelerated from 4 r.p.m. to 40 r.p.m. in 5 min. on day 1. The mice were trained on the beam at 4 r.p.m. constantly until each mouse stayed on the beam for 5 min. On days 2 to 5, the latency was recorded as same as day 1. Three repeats were performed in 1 day with at least 15-min break. For the disease progression experiments[53], the motor performance of mice was recorded in 3 consecutive days, three repeats each day. The average of nine values was recorded for statistical analysis.

**Balance beam test.** The balance beam apparatus (SANS Biological Technology) includes a 50 × 5 cm (length × width) beam that is 50 cm above the floor, and a dark goal box (10 × 10 × 10 cm) at one end of the beam. On day 1, each mouse was trained until running through the beam without pausing. On day 2, the process of each mouse passing through the beam was recorded by video camera. The number of hind-limb slips of each mouse was calculated for further analyses.

**Footprint test.** The footprint test was performed in an apparatus constructed manually[54]. A 60 × 5 × 14 cm (length × width × height) corridor with a dark goal box at the end of the corridor was used. First, the mouse was allowed to habituate to the apparatus for 5 min prior to training. The mouse was then restrained by the scruff of the neck for three times to reduce general anxiety. After the mouse successfully passing through the corridor in the training procedure, a clean sheet of paper was placed on the corridor floor. Fore paws of mice were painted with red dye and hind paws with blue dye. The gait of mice was then recorded in formal tests. The center of each plantar was marked, the distance between the fore paw and hind paw at the same side was measured as back–front distance, and the distance between hind paws at the same side was measured as back stride distance.

**Open-field test.** The open field test was performed in a 40 × 40 × 40 cm (width × length × height) square box (Xinruan Informatlon Technology Co). A 20 × 20 cm central region of the box was marked by Anymaze software (Stoelting Co). The mouse was placed in the corner of the box and recorded by an overhead video camera for 10 min. Several measures including total distance traveled, mean motor speed, and center/total ratio were analyzed.

**Statistical analysis.** Statistical significant was performed using two-tailed, unpaired Student's $t$ tests or two-way ANOVA followed by Bonferroni post hoc test. Two-sided Mann–Whitney testing was used if the data were non-normally distributed. Survival of mice was measured by Kaplan–Meier method, and the difference between two groups was analyzed using Gehan–Breslow–Wilcoxon test. Unless otherwise stated, experiments were performed with male mice in a C57BL/6 genetic background. The data are displayed as mean ± s.e.m. and analyzed using GraphPad Prism 7.00 (GraphPad Software). And $P < 0.05$ were thought as significant difference. *$P < 0.05$, **$P < 0.01$, ***$P < 0.001$, ****$P < 0.0001$.

**Reporting summary.** Further information on research design is available in the Nature Research Reporting Summary linked to this article.

## Data availability

The data of Figs. 1e, 1g, 2b, 2c, 3b, 3c, 3d, 3e, 3g, 3h, 3i, 4a, 4b, 4c, 4d, 4e, 5b, 5d, 5f, 5h, 6c, 6d, 6f, 7e, 7f, and Supplementary Figs. 2b, 2c, 2e, 2f, 2g, 2h, 3a, 3b, 3c, 4b, 4d, 4e, 5a, 5f, 5g, 6i are provided as a Source Data file. The authors will make all data available to readers upon reasonable request. A reporting summary for this article is available as a Supplementary Information file. The RNA sequencing data of this article have been deposited in the NCBI GEO database under accession number GSE132108.

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

## Acknowledgements
We thank Youqiong Ye from the University of Texas Health Science Center at Houston-McGovern Medical School for her assistance with bioinformatics analyses. This work was supported by the National Natural Science Foundation of China (Nos. 81761148024 and 31871023), the National Key Scientific R&D Program of China (No. 2016YFC1306000), Suzhou Clinical Research Center of Neurological Disease (No. Szzx201503) and a Project Funded by the Priority Academic Program Development of Jiangsu Higher Education Institutions.

## Author contributions
Z.H. and G.W. designed experiments; Z.T. constructed the GFP-PR$_{28}$ transgenic mice and plasmids; Z.H., L.L., H.S., Z.Z. and Z.L. performed transgenic mice genotyping, behavioral tests, and qRT-PCR; L.L. and R.W. contributed to immunohistochemical experiments; C.M. and H.R. isolated the RNA and assisted with RNA-sequencing analyses; J.Z. provided Thy1-Cre transgenic mice; G.W. and Z.H. analyzed the data and wrote the paper.

## Additional information

**Competing interests:** The authors declare no competing interests.

