## [Peer Review File · Nature Communications]

Reviewers' comments:

Reviewer #1 (Remarks to the Author):

The authors generated a conditional transgenic line expressing C9orf72 poly-PR proteins (Rosa26 floxed GFP-PR28 x Thy1-Cre). Nucleolar poly-PR is highly toxic in several model systems but poly-PR is extremely rare in C9orf72 patients and was not found in the nucleolus in patients. Homozygous mice die at 1-2 months of age and heterozygous mice seem to develop a motor-phenotype and brain atrophy including the motor cortex and loss of lower motoneurons and Purkinje cells. Microglia activation is surprisingly only found in cerebellum and spinal cord. The mouse model is interesting, but does not provide the promised evidence for the connection of poly-PR and TDP-43 pathology, because "Poly-PR cannot cause cellular inclusion of TDP-43 in short repeat length in vitro and in vivo" (legend Figure S5). The link to synaptic dysfunction is also overstated, because synaptic function was not directly addressed in any experiment. Use of non-littermate controls precludes the analysis of behavioral data. Thus, in the present form the manuscript does not provide novel insights into poly-PR biology in C9orf72 disease.

-Fig 1: The cytoplasmic localization of poly-PR needs to be better characterized since nucleolar poly-PR is probably far more toxic than the cytoplasmic poly-PR found in patients. Fig. 1c shows impressively large aggregates, but it is unclear whether these are remnants of dead cells or truly cytoplasmic. Co-staining with nucleolar and cytoplasmic markers is required.

-Fig 1 and 2: It is surprising that an assumed 2-fold increase in expression changes the lifespan from normal to severely shortened. Expression pattern of the homozygous mice needs to be characterized as well to investigate whether the much stronger phenotype is due to higher expression (needs quantification) or different expression pattern. Is there detrimental off-target expression in other organs?

-Fig S1d. It is surprising that the GFP-PR expression does not fully match Cre expression. The authors should show co-staining of GFP-PR and Cre in cerebellum, hippocampus and spinal cord. It is unclear why poly-PR is most abundant in the granular layer of the cerebellum in humans but not found in this mouse model in these cells despite apparent Cre expression.

-The authors often refer to age-matched controls (line 104, 107, 119, 154, 214, 258, 284, 443, 448, 453). Especially for behavioral analysis (e.g. Fig. 3c-h) this is not acceptable due to effects of maternal care and genetic drift. Without using wildtype littermates as control the results cannot be interpreted.

-The claim of cortical thinning and lack of microglia and astrocyte activation are inconsistent and need to be further explored.

-Fig 6: To exclude that the gene expression changes reflect neuron loss the analysis should be normalized to neuron-specific genes.

-Fig 6b, S4c: The gene names need to be included to allow meaningful conclusions.

-Fig 7: This figure is highly over-interpreted and misleading. Loss of TDP-43 is highly neurotoxic and thus the gene expression changes in TDP-43 knockdown and PR expression mice may just reflect neurodegeneration rather than specific TDP-43 effects. E.g. upregulation of C1qa, CD68 and TREM2 is most likely due to a microglia response to neuronal debris and not a direct effect of TDP-43 loss of function. Differential splicing of validated TDP-43 targets or cryptic exons should be analyzed instead. In fact, the lack of expression changes of TDP-43 and lack of TDP-43 mislocalization strongly argue against direct TDP-43 effects (Fig 7e, S5). Moreover, effects of poly-PR on TDP-43 were only shown for much longer poly-PR constructs in vitro (Fig 7f). The co-immunoprecipitation of TDP-43 and poly-PR should be repeated directly in mouse tissue.

Reviewer #2 (Remarks to the Author):

Hao et al provide the first description of transgenic mice expressing polyPR repeats. They show that homozygous mice expressing 28 polyPR repeats have a early and severe phenotypes including early death so concentrate on the heterozygous mice. The heterozygous mice have polyPR inclusions in several brain areas and have impaired motor function at 6 months of age. Evidence is also provided for neuronal loss at 6 months of age and an increase in astrogliosis and microgliosis. RNA-seq of the cerebellum suggested synaptic dysfunction. Preliminary data suggested an interaction of TDP-43 and polyPR and that longer polyPR repeats can induce cytoplasmic TDP-43 aggregates in cell culture.

Overall the manuscript will be of interest as it is the first polyPR mouse. However, further information is needed to better understand the key features of the model. I am also concerned about the length of the polyPR repeats, as 28 repeats seems rather short as compared to the likely length in patients. The authors own cell culture data (Figure 7) suggests that longer repeats may provide a better model. One reason for this is that it is likely that most of the polyPR inclusions are intranuclear/nucleolar, which is not observed in patients. This is not necessarily a major concern, as it could represent an early disease stage and all models have limitations, but it is important to determine the localisation of the polyPR inclusions and whether their location changes over time. Points to address:

1. Figure 1. Provide data on the localisation of the polyPR inclusions, in comparison to a nucleolar marker, with quantification, to show where they are predominantly located and the burden of neurons that have inclusions, in important brain areas.
2. Figure 1. It is important to determine whether the heterozygotes have a progressive phenotype, therefore quantification of inclusions, as above, at a second time point would be very insightful.
3. Figure 2. It would be helpful to provide qPCR and western blots to compare polyPR levels in the homozygous and heterozygous mice.
4. Figures 2 and 3. The sex of the mice are not specified. This is very important to state, especially for the weight and grip strength data. The sexes should be analysed separately for these analyses.
5. Figure 2 and S2. Was the grip strength data normalised for body weight? Please also provide the normalised data to help determine whether body weight differences could be driving the effect.
6. Figure 4. It is important to know whether the decreased neuron counts are developmental or due to a degenerative effect of polyPR. The neuronal counts should be repeated at an earlier timepoint such as 2 months and if possible also a later timepoint. Please state the age of the mice for panel 4 d and e.
6. Figure 4. More detail is required on the methods for neuronal counting. For motor neuron counts, how many sections were counted per mouse and how were sections matched to ensure the same level of spinal cord were assessed in each mouse. It is essential that sections were carefully matched and more than one section was counted per mouse. The number of sections counted and how neuroanatomical matching of sections was performed is also needed for the Purkinje cell counts and cortical thickness measurements.
7. Figure 5. More information is needed on how GFAP and Iba1 measurements were performed – how were sections matched, was a defined area measured in each section? How many sections were measured per mouse?
8. Figure 5. An additional earlier timepoint would be helpful to determine whether the gliosis is progressive.
9. Figure 6 is difficult to interpret as the changes observed could be due to loss of Purkinje neurons rather than a specific effect on synaptic transmission. The interpretation of this figure should be toned down accordingly.
10. Figure 7. Please provide the number of biological replicates performed for 7d and 7f. Quantification of intranuclear and cytoplasmic inclusions of TDP-43 is required for 7f.
11. Fig S5a. The TDP-43 staining does not look like the characteristic nuclear staining, in controls or heterozygous mice. The TDP-43 antibody used is also not stated. This should be repeated with optimised TDP-43 staining.

Reviewer #3 (Remarks to the Author):

Hexanucleotide GGGGCC repeat expansion in C9ORF72 is the most common genetic cause of ALS and FTD. Five dipeptide repeat proteins (DPRs) have been shown to be produced from both sense and antisense RNA repeats by repeat-associated non-ATG (RAN) translation. Arginine-containing ones, poly-GR and poly-PR were reported to be most toxic by many previous studies, using cultured cells, yeast, *C.elegans* and *Drosophila* models, as well as a recent publication on toxicity of poly-GR (GR100) in mice. This study reported a new transgenic mouse line expressing GFP-PR28 specifically in neurons. They found the homozygous mice have decreased survival, while the heterozygous mice showed phenotypes related to motor neuron and Purkinje cell dysfunction. Transcriptome analysis identified altered gene expression linked to synaptic dysfunction in cerebellum, and it is claimed to be correlated with changes in TDP-43 knockdown neurons. This study provides a new mouse model and research tool to understand the pathogenesis of poly-PR. However, most of the work is descriptive and not mechanistic. And there are some designing weakness that make the disease relevance of this work less appealing.

Major concerns:

1. This biggest problem is the poly-PR only has 28 repeats. The cut-off to discriminate between normal repeat alleles and pathogenic expanded repeats is generally believed to be around 24-35 repeats, which slightly varies in different studies. Whether 28 repeats of GGGGCC is pathogenic is not clear. Although it is hard to engineer very long repeats due to technical difficulties, 28-repeat is still too short for disease relevance. Furthermore, RAN translation efficiency from short repeats is significantly lower than longer repeats, producing lower levels of DPRs if any. Therefore, massive overexpression of very short repeats of PR dipeptide might not recapitulate the disease condition at all.
2. The number of animals in each group is too low. The authors only used around 7 mice per genotype to compare the behavior defects between control and PR28. For behavior test, there should be at least around 15 animals per group to have statistic significance.
3. The timeline of the disease course is not clear. Many of the figures and text didn't describe the age of the animals. Does behavior abnormalities correlate with pathology? Many of the behavior tests were only performed at one time point (such as footprint, open field, cage behavior, balance beam). When did the phenotype begin to appear? The authors showed brain weight and cerebellum weight were already reduced at 2-month old (Fig b,c). This shows that these are developmental problems but not degenerate phenotypes. In addition, the body weight decrease was only observed in male heterozygous mice. What about other behavior tests? Are there gender differences?
4. Figure 1: The percentage of different types of neurons expressing the PR28 should be quantified. The sub-cellular localization of PR aggregates is not clear. It is claimed to be nuclear aggregates. But in several brain regions, it seems to be cytosol aggregates? In 1d, why chat staining showed punta?
5. RNA-seq was only performed on cerebellum. It is more important to examine motor cortex and spinal cord, as these regions are more relevant to human disease.
6. The authors found the transcriptome changes in the cerebellum of PR28 mice correlate with ones caused by TDP-43 reduction. They hypothesized that poly-PR might sequester TDP-43 and leads to its loss of function. However, no TDP-43 inclusions were found in the PR28 mice, even at 12-month old (the RNA-seq was done using 5-month animal). The authors therefore tested whether there is length dependence, using cell cultures. The cytosol aggregation of TDP-43 was only observed in cells expressing long poly-PR (PR96). This evidence actually argues against the interaction between PR28 and TDP-43 in mice. Therefore, the overlapped gene changes in mice (which only has 28 repeats) might just be a coincidence of cell death or dysfunction, but not a mechanistic linkage. More importantly, this also indicates the mouse model with very short repeats might have intrinsic problems to dissect pathological pathways in human disease, as the longer repeats probably have different features.

Minor points:

1. The relative expression levels of PR28 in heterozygous versus homozygous mice should be quantified.

2. Fig.S5a: Immunohistochemical staining of TDP-43 should be performed in the cerebellum.
3. In Discussion on page 9: it is not proper to compared toxicity of PR with the AAV-GA or GR mice, as these are different strategies and the relative expression levels of these DPRs are not known.
4. In Result, first sentence: ploy-PR should be poly-PR.

We are returning the revised manuscript entitled “Motor dysfunction and neurodegeneration in a C9orf72 mouse line expressing poly-PR” (NCOMMS-18-28985). We thank Reviewers for all the constructive and detailed suggestions that are of help for us to improve our manuscript. In the revised manuscript, we increased animal numbers, performed extra experiments and added the new data following the suggestions by Reviewers. We also modified the manuscript as suggested.

We do thank all Reviewers for the constructive suggestions, especially in the characterization of progressive changes in heterozygotes, both pathology and behaviors. It is absolutely helpful for improvement of manuscript.

The followings are our point-to-point responses to the concerns raised by Reviewers.

Reviewer #1 (Remarks to the Author):

The authors generated a conditional transgenic line expressing C9orf72 poly-PR proteins (Rosa26 floxed GFP-PR₂₈ x Thy1-Cre). Nucleolar poly-PR is highly toxic in several model systems but poly-PR is extremely rare in C9orf72 patients and was not found in the nucleolus in patients. Homozygous mice die at 1-2 months of age and heterozygous mice seem to develop a motor-phenotype and brain atrophy including the motor cortex and loss of lower motoneurons and Purkinje cells. Microglia activation is surprisingly only found in cerebellum and spinal cord. The mouse model is interesting, but does not provide the promised evidence for the connection of poly-PR and TDP-43 pathology, because "Poly-PR cannot cause cellular inclusion of TDP-43 in short repeat length in vitro and in vivo" (legend Figure S5). The link to synaptic dysfunction is also overstated, because synaptic function was not directly addressed in any experiment. Use of non-littermate controls precludes the analysis of behavioral data. Thus, in the present form the manuscript does not provide novel insights into poly-PR biology in C9orf72 disease.

Point 1

Fig 1: The cytoplasmic localization of poly-PR needs to be better characterized since nucleolar poly-PR is probably far more toxic than the cytoplasmic poly-PR found in patients. Fig. 1c shows impressively large aggregates, but it is unclear whether these are remnants of dead cells or truly cytoplasmic. Co-staining with nucleolar and cytoplasmic markers is required.

Response

We thank Reviewer 1 for suggesting us to clarify the poly-PR subcellular localization. We improved the protocol for immunostaining, in which 0.1% sudan black B in 70% ethanol was added to quench the autofluorescence. The cytoplasmic aggregates of poly-PR were rarely

detected, although there was diffused cytoplasmic distribution in some cells, such as motor neurons in spinal cord (Fig. 1). We performed co-staining with nucleolar marker (Nucleolin) (Supplementary Fig. 1a, b) as suggested by Reviewer 1. After quenching the autofluorescence, it became clearer that the poly-PR was nuclear (indicated by DAPI staining), and the aggregates were nucleolar. We do thank Reviewer 1 for this point.

Point 2

Fig 1 and 2: It is surprising that an assumed 2-fold increase in expression changes the lifespan from normal to severely shortened. Expression pattern of the homozygous mice needs to be characterized as well to investigate whether the much stronger phenotype is due to higher expression (needs quantification) or different expression pattern. Is there detrimental off-target expression in other organs?

Response

We appreciate Reviewer 1 for this detailed comments and suggestions. In Fig. 1e of our revised manuscript, the relative expression levels of GFP in control, heterozygous and homozygous mice were determined, which showed about two-fold increases of GFP (reflects GFP-poly-PR) in homozygotes as compared to heterozygotes, but no expression of GFP in control groups. In addition, the expressions of GFP were not detected in peripheral tissues in either homozygotes or heterozygotes.

Point 3

Fig S1d. It is surprising that the GFP-PR expression does not fully match Cre expression. The authors should show co-staining of GFP-PR and Cre in cerebellum, hippocampus and spinal cord. It is unclear why poly-PR is most abundant in the granular layer of the cerebellum in humans but not found in this mouse model in these cells despite apparent Cre expression.

Response

We followed the suggestions by Reviewer 1 and performed extra experiments. In Supplementary Fig. 1d in our revised manuscript, co-staining of GFP-PR₂₈ and Cre in cerebellum, hippocampus and spinal cord was performed. Co-localization/distribution of GFP and Cre was found in hippocampus and spinal cord. However, GFP-PR₂₈ was not expressed in the granular layer of the cerebellum, where Cre was well expressed. It is really surprising that the GFP-PR₂₈ expression did not fully match Cre expression. One possibility is that the expression of poly-PR in the granular layer of the cerebellum in our model was too low.

Point 4

The authors often refer to age-matched controls (line 104, 107, 119, 154, 214, 258, 284, 443, 448, 453). Especially for behavioral analysis (e.g. Fig. 3c-h) this is not acceptable due to effects of maternal care and genetic drift. Without using wildtype littermates as control the results cannot be interpreted.

Response

We apologize that we did not describe clearly in our manuscript in which we used

“age-matched control”, but they were littermates. The breeding strategy that we used was that we first crossed the GFP-PR₂₈^{flox/+} mice to get GFP-PR₂₈^{flox/flox} mice, and used male GFP-PR₂₈^{flox/flox} mice to cross with female GFP-PR₂₈^{flox/flox} mice to expand GFP-PR₂₈^{flox/flox} mouse number. If take a look at Fig. 2b, which shows that we crossed the GFP-PR₂₈^{flox/flox} mice with the Thy1-Cre^{+/-} mice so that we got GFP-PR₂₈^{flox/+}; Thy1-Cre^{+/-} mice and GFP-PR₂₈^{flox/+}; Thy1-Cre^{-/-} mice. By this way, we got the heterozygotes GFP-PR₂₈^{flox/+} with Cre (Thy1-Cre^{+/-}) and without Cre (Thy1-Cre^{-/-}). They are littermates. The GFP-PR₂₈^{flox/+}; Thy1-Cre^{-/-} mice are used as controls. For producing the homozygotes with Cre, we crossed GFP-PR₂₈^{flox/flox} mice with GFP-PR₂₈^{flox/+}; Thy1-Cre^{+/-} mice. So we got the mice with following genotypes: GFP-PR₂₈^{flox/+}; Thy1-Cre^{+/-}, GFP-PR₂₈^{flox/+}; Thy1-Cre^{-/-}, GFP-PR₂₈^{flox/flox}; Thy1-Cre^{+/-}, GFP-PR₂₈^{flox/flox}; Thy1-Cre^{-/-}. The GFP-PR₂₈^{flox/flox}; Thy1-Cre^{-/-} was used as control mice of GFP-PR₂₈^{flox/flox}; Thy1-Cre^{+/-} transgenic mice. The description was added in our revised manuscript.

Point 5

The claim of cortical thinning and lack of microglia and astrocyte activation are inconsistent and need to be further explored.

Response

We appreciate Reviewer 1 for this suggestion. The activation of astrocytes was further examined. Unfortunately, we did not found a significant activation of astrocytes (Supplementary Fig. 3f) in the motor cortex of heterozygous mice, although the reason is still unclear.

Point 6

Fig 6: To exclude that the gene expression changes reflect neuron loss the analysis should be normalized to neuron-specific genes.

Response

We followed the suggestion by Reviewer 1. The relative expression levels of synaptic related genes were normalized to Purkinje specific expressed gene Pcp2 (Fig. 6e). The expression levels of synaptic related genes were still significantly decreased.

Point 7

Fig 6b, S4c: The gene names need to be included to allow meaningful conclusions.

Response

Thank Reviewer 1 for this suggestion. In our revised manuscript, the representative top ten genes changed were presented in right panel of heatmap (Supplementary Fig. 4c).

Point 8

Fig 7: This figure is highly over-interpreted and misleading. Loss of TDP-43 is highly neurotoxic and thus the gene expression changes in TDP-43 knockdown and PR expression mice may just reflect neurodegeneration rather than specific TDP-43 effects. E.g.

upregulation of C1qa, CD68 and TREM2 is most likely due to a microglia response to neuronal debris and not a direct effect of TDP-43 loss of function. Differential splicing of validated TDP-43 targets or cryptic exons should be analyzed instead. In fact, the lack of expression changes of TDP-43 and lack of TDP-43 mislocalization strongly argue against direct TDP-43 effects (Fig 7e, S5). Moreover, effects of poly-PR on TDP-43 were only shown for much longer poly-PR constructs in vitro (Fig 7f). The co-immunoprecipiation of TDP-43 and poly-PR should be repeated directly in mouse tissue.

Response

We greatly thank Reviewer 1 for the critical comments and suggestions. To further confirm the involvement of TDP-43 in our animal model, we examined alternative splicing of TDP-43 targets using RNAs extracted from the cerebellum of 5 months old control and heterozygous mice, with primers elsewhere (Lagier-Tourenne et al., 2012). Unfortunately, no significant difference between two groups was observed, suggesting that the dysregulated genes in poly-PR expressed cerebellum might be not due to TDP-43 loss of function. So we agree with Reviewer 1, which the changes may be caused by neurodegeneration, but not the effects of TDP-43. We corrected the overstatement of role of TDP-43 and deleted this figure in the revised manuscript.

Here, we thank all three Reviewers pointing out the defects of TDP-43 data (Figure 7). It may also like Reviewer 3 pointed “the overlapped gene changes in mice might just be a coincidence of cell death or dysfunction, but not a mechanistic linkage”. We now recognized that it is really no “mechanistic linkage”, so that we deleted this figure.

Reviewer #2 (Remarks to the Author):

Hao et al provide the first description of transgenic mice expressing polyPR repeats. They show that homozygous mice expressing 28 polyPR repeats have a early and severe phenotypes including early death so concentrate on the heterozygous mice. The heterozygous mice have polyPR inclusions in several brain areas and have impaired motor function at 6 months of age. Evidence is also provided for neuronal loss at 6 months of age and an increase in astrogliosis and microgliosis. RNA-seq of the cerebellum suggested synaptic dysfunction. Preliminary data suggested an interaction of TDP-43 and polyPR and that longer polyPR repeats can induce cytoplasmic TDP-43 aggregates in cell culture.

Overall the manuscript will be of interest as it is the first polyPR mouse. However, further information is needed to better understand the key features of the model. I am also concerned

about the length of the polyPR repeats, as 28 repeats seems rather short as compared to the likely length in patients. The authors own cell culture data (Figure 7) suggests that longer repeats may provide a better model. One reason for this is that it is likely that most of the polyPR inclusions are intranuclear/nucleolar, which is not observed in patients. This is not necessarily a major concern, as it could represent an early disease stage and all models have limitations, but it is important to determine the localisation of the polyPR inclusions and whether their location changes over time.

Point 1

Figure 1. Provide data on the localisation of the polyPR inclusions, in comparison to a nucleolar marker, with quantification, to show where they are predominantly located and the burden of neurons that have inclusions, in important brain areas.

Response

We deeply thank Reviewer 2 for pointing this out. Pictures of co-staining of GFP with Neucleolin in major brain regions were presented in our revised manuscript (Supplementary Fig. 1a, b). The poly-PR was major intranuclear localization in neurons in most regions, excepted for in lumbar motor neuron, in which showed intranuclear with diffused cytoplasmic distribution (Fig. 1c). And the percentage of cells with poly-PR inclusions was counted (Fig. 1g). The cerebellar Purkinje cells had the highest expression of poly-PR. Consistently, the Purkinje cells were dramatically damaged.

Point 2

Figure 1. It is important to determine whether the heterozygotes have a progressive phenotype, therefore quantification of inclusions, as above, at a second time point would be very insightful.

Response

We thank Reviewer 2 for this suggestion. The localization of GFP-PR₂₈ was further determined in neurons of cerebellum and cortex of 12 months old heterozygous mice (Fig. 1f). The percentage of cells with GFP positive inclusions remained unchanged in 12 months old heterozygous mice as compared with 2 months old heterozygous mice (Fig. 1g). However, a progressive loss of Purkinje cells from 2 to 12 months old heterozygous mice was observed (Fig. 4g).

Point 3

Figure 2. It would be helpful to provide qPCR and western blots to compare polyPR levels in the homozygous and heterozygous mice.

Response

We followed the suggestion by Reviewer 2 and performed qRT-PCR to determine the expression level of poly-PR. The data were showed in Fig. 1e.

Point 4

Figures 2 and 3. The sex of the mice are not specified. This is very important to state, especially for the weight and grip strength data. The sexes should be analysed separately for these analyses.

Response

We gratefully thank Reviewer 2 for the critical suggestions. We previously used male mice in the study. In our revised manuscript, the motor function of female heterozygous mice was examined. Consistent with the results from male heterozygous mice, the female mice also developed hindlimbs clasping, age-dependent decreased body weight, decreased latency on the rotarod. However, no significant difference of grip strength was found in male and female mice (Supplementary Fig. 2d-h).

Point 5

Figure 2 and S2. Was the grip strength data normalised for body weight? Please also provide the normalised data to help determine whether body weight differences could be driving the effect.

Response

Thank you for this important suggestion. The normalized data of grip strength were showed below. After normalization with body weight, no significant difference between control and homozygous mice at 20 days of age were observed (a), so we removed these data in our revised manuscript. Interestingly, the normalized grip strength of male heterozygous mice showed slight stronger than control mice (b, c). As the heterozygous mice showed obvious anxiety (Fig. 3k, l), they grasped the grid tightly in grip strength test. Due to more severe in decrease of body weight, loss of body weight greatly contributed to the increase of the grip strength of the heterozygous mice

Point 6

Figure 4. It is important to know whether the decreased neuron counts are developmental or due to a degenerative effect of polyPR. The neuronal counts should be repeated at an earlier timepoint such as 2 months and if possible also a later timepoint. Please state the age of the mice for panel 4 d and e.

Response

We thank Reviewer 2 for the critical comments and suggestion. The numbers of neuron of 2 and 12 months old control and heterozygous mice were counted (Fig. 4). No neuronal loss was observed in Purkinje cells and lumbar motor neurons in 2 months old heterozygous mice (Fig. 4g and 4k). But the 2 months old heterozygous mice showed atrophy of cerebellar molecular layer. It is possibility that poly-PR induces neurite degeneration before neuronal loss in cerebellum. However, more severe loss of Purkinje cells at 12 months old heterozygous mice than 6 months old heterozygous mice was observed (Fig. 4g), which may further suggest an involvement of neurodegeneration rather than development.

Point 7

More detail is required on the methods for neuronal counting. For motor neuron counts, how many sections were counted per mouse and how were sections matched to ensure the same level of spinal cord were assessed in each mouse. It is essential that sections were carefully matched and more than one section was counted per mouse. The number of sections counted and how neuroanatomical matching of sections was performed is also needed for the Purkinje cell counts and cortical thickness measurements.

Response

In our revised manuscript, more detail on the methods of neuronal counting was added. Sections were carefully matched and average 3 sections per mouse, 3-5 mice per group were used. For lumbar motor neurons counting, nine sections per mouse, 3-5 mice per group were used.

Point 8

Figure 5. More information is needed on how GFAP and Iba1 measurements were performed – how were sections matched, was a defined area measured in each section? How many sections were measured per mouse?

Response

Similar with the methods of neuronal counting, the methods for GFAP and Iba1 measurements were also described in “Quantification” of Materials and methods section in the revision (please also refer to response to point 7).

Point 9

Figure 5. An additional earlier timepoint would be helpful to determine whether the gliosis is progressive.

Response

Thank Reviewer 2 for this suggestion. The activation of glia in 2 months old control and heterozygous mice was presented (Fig. 5c, d, f) in the revision. Gliosis were not observed in cerebellum of 2 months old heterozygous mice, while glia cells were activated in 6 months old mice, suggesting a progressive phenotype.

Point 10

Figure 6 is difficult to interpret as the changes observed could be due to loss of Purkinje neurons rather than a specific effect on synaptic transmission. The interpretation of this figure should be toned down accordingly.

Response

Thank you for this suggestion. We modified the interpretation following the suggestion.

Point 11

Figure 7. Please provide the number of biological replicates performed for 7d and 7f. Quantification of intranuclear and cytoplasmic inclusions of TDP-43 is required for 7f.

Response

We thank all three Reviewers pointing out the defects of TDP-43 data (Figure 7). The aggregates of TDP-43 occurred in cells with high expression of long poly-PR (96 repeats) but not short repeats *in vitro*, which is repeatable. However, we could not find the inclusions in animals, although we carefully checked the TDP-43 inclusions in animals at different ages. Consistent with the data “no aggregates of TDP-43 were presented in our poly-PR transgenic mice” in the original manuscript. It may just like Reviewer 1 pointed “over-interpreted and misleading” (Reviewer 1, point 8) and also like Reviewer 3 pointed “the overlapped gene changes in mice might just be a coincidence of cell death or dysfunction, but not a mechanistic linkage”. Thus, we deleted this figure.

Point 12

Fig S5a. The TDP-43 staining does not look like the characteristic nuclear staining, in controls or heterozygous mice. The TDP-43 antibody used is also not stated. This should be repeated with optimised TDP-43 staining.

Response

Optimized TDP-43 staining was added in Supplementary Fig. 3c, and the information of TDP-43 antibody is now added in “Immunohistochemical analysis” of Materials and methods.

Reviewer #3 (Remarks to the Author):

Hexanucleotide GGGGCC repeat expansion in C9ORF72 is the most common genetic cause of ALS and FTD. Five dipeptide repeat proteins (DPRs) have been shown to be produced from both sense and antisense RNA repeats by repeat-associated non-ATG (RAN) translation. Arginine-containing ones, poly-GR and poly-PR were reported to be most toxic by many previous studies, using cultured cells, yeast, *C.elegans* and *Drosophila* models, as well as a recent publication on toxicity of poly-GR (GR100) in mice. This study reported a new transgenic mouse line expressing GFP-PR28 specifically in neurons. They found the

homozygous mice have decreased survival, while the heterozygous mice showed phenotypes related to motor neuron and Purkinje cell dysfunction. Transcriptome analysis identified altered gene expression linked to synaptic dysfunction in cerebellum, and it is claimed to be correlated with changes in TDP-43 knockdown neurons. This study provides a new mouse model and research tool to understand the pathogenesis of poly-PR. However, most of the work is descriptive and not mechanistic. And there are some designing weakness that make the disease relevance of this work less appealing.

Point 1

This biggest problem is the poly-PR only has 28 repeats. The cut-off to discriminate between normal repeat alleles and pathogenic expanded repeats is generally believed to be around 24-35 repeats, which slightly varies in different studies. Whether 28 repeats of GGGGCC is pathogenic is not clear. Although it is hard to engineer very long repeats due to technical difficulties, 28-repeat is still too short for disease relevance. Furthermore, RAN translation efficiency from short repeats is significantly lower than longer repeats, producing lower levels of DPRs if any. Therefore, massive overexpression of very short repeats of PR dipeptide might not recapitulate the disease condition at all.

Response

We thank Reviewer 3 for the critical comments. We agree with Reviewer 3 for that 28-repeats are short. But it was hard to choose the length of repeats for animal preparation due to no published reference for poly-PR transgenic mice. In drosophila, 36-repeats of PR or GR causes eye degeneration and lethality (Mizielinska et al., 2014). The transgenic drosophila with 50-repeats of PR produce severe neurodegeneration (almost loss of all eye neurons, leading to undetectable of poly-PR by western blot) and can not develop to adulthood (Wen et al., 2014). Moreover, 25 repeats of poly-PR have significant cellular toxicity for primary neurons (Wen et al., 2014). In our cellular model, poly-PR is the most toxic among the five DPRs, and 28-repeats of PR sufficiently induced primary neuronal death (unpublished data). Thus, we carefully chose 28-repeats for animal preparation to avoid too high toxicity for animal survival. But we do agree with Reviewer 3 that it is possible that longer repeats may be better.

Point 2

The number of animals in each group is too low. The authors only used around 7 mice per genotype to compare the behavior defects between control and PR28. For behavior test, there should be at least around 15 animals per group to have statistic significance.

Response

Thank you for this suggestion. We followed the suggestion and re-performed behavior test using more mice in behavior tests. But we apologize that we have no enough animals more than 12 months old, the number of old group is lower than 15.

Point 3

The timeline of the disease course is not clear. Many of the figures and text didn't describe the

age of the animals. Does behavior abnormalities correlate with pathology? Many of the behavior tests were only performed at one time point (such as footprint, open field, cage behavior, balance beam). When did the phenotype begin to appear? The authors showed brain weight and cerebellum weight were already reduced at 2-month old (Fig b,c). This shows that these are developmental problems but not degenerate phenotypes. In addition, the body weight decrease was only observed in male heterozygous mice. What about other behavior tests? Are there gender differences?

Response

Thank Reviewer 3 for these critical suggestions, which are also pointed by Reviewer 2. The disease progression of heterozygous mice was performed with 3, 6, 10 and 12-16 months old mice using rotarod test (Fig. 3m). And the disease progression was summarized in Fig. 3o. Briefly, the heterozygous mice developed hindlimbs claspings at 2 months of age, but without motor dysfunction determined using rotarod. The motor dysfunction initiated at about 6 months of age, with reduced numbers of Purkinje cell and lumbar motor neuron, and motor dysfunction of mice became more severe at 10 months of age. Finally, the mice showed decreased body weight and dramatic decreased survival at 12 months of age. Six and 12 months old mice were also subjected to footprint test, and the data were added in the revised manuscript.

The early loss of brain weight and cerebellum weight may be caused by high expression of poly-PR, which is similar to the findings in the recent published GR100 transgenic mice (Zhang et al., 2018).

The data of female mice was added in Supplementary Fig. 2f in our revised manuscript. Tail suspension test and rotarod test were performed in female mice too (Supplementary Fig. 2e, g). Similar to male mice, the female mice also developed motor phenotypes.

Point 4

Figure 1: The percentage of different types of neurons expressing the PR28 should be quantified. The sub-cellular localization of PR aggregates is not clear. It is claimed to be nuclear aggregates. But in several brain regions, it seems to be cytosol aggregates? In 1d, why chat staining showed punta?

Response

We thank Reviewer 3 for this comment that was also pointed out by Reviewer 1 and 2. The percentage of cells with poly-PR aggregates in major brain regions was quantified in Fig. 1g. The cellular localization of poly-PR in major brain regions was examined using improved immunostaining protocol, by which the slides were treated with 0.1% Sudan Black B to quench the autofluorescence. After quenching the autofluorescence, no cytoplasmic puncta were observed.

Point 5

RNA-seq was only performed on cerebellum. It is more important to examine motor cortex and spinal cord, as these regions are more relevant to human disease.

Response

Thank you for this suggestion, the RNA-seq results of motor cortex and lumbar spinal cord were showed in our revised manuscript (Supplementary Fig. 5). Gene Ontology analyses of enriched categories identified 'Positive regulation of neurotransmitter secretion' and 'Exocytosis' were major pathway implicated in dysregulated genes of cortex, which is consistent with the results of cerebellum. Moreover, inflammation related genes were both upregulated in cerebellum and spinal cord of heterozygous mice.

Point 6

The authors found the transcriptome changes in the cerebellum of PR28 mice correlate with ones caused by TDP-43 reduction. They hypothesized that poly-PR might sequester TDP-43 and leads to its loss of function. However, no TDP-43 inclusions were found in the PR28 mice, even at 12-month old (the RNA-seq was done using 5-month animal). The authors therefore tested whether there is length dependence, using cell cultures. The cytosol aggregation of TDP-43 was only observed in cells expressing long poly-PR (PR96). This evidence actually argues against the interaction between PR28 and TDP-43 in mice. Therefore, the overlapped gene changes in mice (which only has 28 repeats) might just be a coincidence of cell death or dysfunction, but not a mechanistic linkage. More importantly, this also indicates the mouse model with very short repeats might have intrinsic problems to dissect pathological pathways in human disease, as the longer repeats probably have different features.

Response

We gratefully thank Reviewer 3 for the critical comments and suggestions. This picture was also questioned by Reviewer 1 and reviewer 2. We examined alternative splicing of TDP-43 targets using RNAs extracted from the cerebellum of 5 months old control and heterozygous mice, with primers elsewhere (Lagier-Tourenne et al., 2012). Unfortunately, no significant difference between two groups was observed, suggesting that the dysregulated genes in poly-PR expressed cerebellum might be not due to TDP-43 loss of function. So we agree with all Reviewers, which the changes may be caused by neurodegeneration or just be a coincidence of cell death or dysfunction, but not a mechanistic linkage to TDP-43. Thus, we deleted this figure in the revised manuscript. Please also refer to response to Reviewer 1, point 1.

Point 7

The relative expression levels of PR28 in heterozygous versus homozygous mice should be quantified.

Response

We thank Reviewer 3 for this suggestion that was also pointed out by Reviewer 2. In Fig. 1e of our revised manuscript, the relative expression levels of GFP-PR₂₈ in heterozygous and homozygous mice were determined using qRT-PCR. About two-fold increase in the expression of GFP was observed in the homozygotes as compared to the heterozygotes.

Point 8

Fig.S5a: Immunohistochemical staining of TDP-43 should be performed in the cerebellum.

Response

Immunohistochemical staining of TDP-43 in the cerebellum and spinal cord was added in Supplementary Fig. 3c of our revised manuscript.

Point 9

In Discussion on page 9: it is not proper to compare toxicity of PR with the AAV-GA or GR mice, as these are different strategies and the relative expression levels of these DPRs are not known.

Response

We thank Reviewer 3 for this suggestion. It is really incomparable as these are different strategies and the relative expression levels for different DPRs in mice. We deleted this part in our revised manuscript.

Point 10

In Result, first sentence: ploy-PR should be poly-PR.

Response

We apologize for our neglects. It has been corrected in our revised manuscript.

Finally, we thank all Reviewers again for reviewing our manuscript and their suggestions.
Sincerely,

Guanghai Wang

References

- Lagier-Tourenne, C., Polymenidou, M., Hutt, K.R., Vu, A.Q., Baughn, M., Huelga, S.C., Clutario, K.M., Ling, S.C., Liang, T.Y., Mazur, C., *et al.* (2012). Divergent roles of ALS-linked proteins FUS/TLS and TDP-43 intersect in processing long pre-mRNAs. *Nat Neurosci* 15, 1488-1497.
- Mizielinska, S., Gronke, S., Niccoli, T., Ridler, C.E., Clayton, E.L., Devoy, A., Moens, T., Norona, F.E., Woollacott, I.O.C., Pietrzyk, J., *et al.* (2014). C9orf72 repeat expansions cause neurodegeneration in *Drosophila* through arginine-rich proteins. *Science* 345, 1192-1194.
- Wen, X., Tan, W., Westergard, T., Krishnamurthy, K., Markandaiah, S.S., Shi, Y., Lin, S., Shneider, N.A., Monaghan, J., Pandey, U.B., *et al.* (2014). Antisense proline-arginine RAN dipeptides linked to C9ORF72-ALS/FTD form toxic nuclear aggregates that initiate *in vitro* and *in vivo* neuronal death. *Neuron* 84, 1213-1225.
- Zhang, Y.J., Gendron, T.F., Ebbert, M.T.W., O'Raw, A.D., Yue, M., Jansen-West, K., Zhang, X., Prudencio, M., Chew, J., Cook, C.N., *et al.* (2018). Poly(GR) impairs protein translation and stress granule dynamics in C9orf72-associated frontotemporal dementia and amyotrophic lateral sclerosis. *Nat Med* 24, 1136-1142.

Reviewers' comments:

Reviewer #1 (Remarks to the Author):

The manuscript greatly improved

Reviewer #2 (Remarks to the Author):

The revised manuscript is now much clearer and shows that overexpressing a short PR peptide can cause neurodegeneration and associated inflammation in mice. The data on synaptic transmission are still overstated. No direct evidence of a synaptic transmission defect is reported. A reduction in synaptic protein RNAs is shown, which could reflect ongoing neurodegeneration.

Other comments:

Fig 1c. the authors state there is diffuse cytoplasmic PR-GFP staining in lumbar motor neurons but no control is included to allow comparison to the level of background staining. Images must be provided of GFP staining in the control sections taken with the same settings as for the PR mice.

Reviewer #3 (Remarks to the Author):

In the revised manuscript about a new C9ORF72-ALS/FTD transgenic mouse model expressing GFP-PR28 in neurons, the data quality has been improved overall. However, even not considering the caveats of the short repeat length, there is still limited novel mechanistic insight from this work. Especially the link with TDP-43 aggregation is now proved to be negative, a concern raised by several reviewers. But there are still many other pathological features related to C9 reported by many groups, including nucleocytoplasmic trafficking defects, autophagy dysfunction, ribosome dysfunction, stress granule, DNA damage, heterochromatin anomalies (recently reported in the AAV-PR50 mice). It is very important to characterize this mouse model more carefully and thoroughly and correlate these pathological features with behavior tests. This can help get a better understanding of the possible underlying mechanisms leading to the phenotypic defects.

In addition, there are several issues with the RNA-seq analysis:

First of all, what's the cutoff of gene expression differences between two groups? Besides P values, the fold change threshold (such as >1.5 or 2 fold difference) also need to be taken into consideration.

In Fig. S5e, it shows there are total 267 gene changes in spinal cord. But in the Venn diagram in S5g, there are only 52 in spinal cord. Why?

In Fig. S4d, what are "interacted genes"?

When comparing the overlaps of the gene changes, including cerebellum vs spinal cord and cerebellum 2-month vs 5-month, the upregulated and downregulated genes need to be compared separately. It is possible the overlapped gene expression changes actually have opposite directions.

An earlier time point of spinal cord RNA-seq should be included and compare with cerebellum. As this manuscript focuses on motor dysfunction, it is important to test whether the synaptic transmission-related genes are also altered in spinal cord neurons.

A list of all the gene expression changes in each tissue and time point should be included in a supplementary table, which contains gene name, ID, q values and fold differences, et al.

We thank all reviewers for careful review and for constructive suggestions. We revised our manuscript as suggested. The followings are responses to Reviewers.

Reviewer #1 (Remarks to the Author):

The manuscript greatly improved

Response

We thank Reviewer 1 for positive comment.

Reviewer #2 (Remarks to the Author):

The revised manuscript is now much clearer and shows that overexpressing a short PR peptide can cause neurodegeneration and associated inflammation in mice. The data on synaptic transmission are still overstated. No direct evidence of a synaptic transmission defect is reported. A reduction in synaptic protein RNAs is shown, which could reflect ongoing neurodegeneration.

Other comments:

Fig 1c. the authors state there is diffuse cytoplasmic PR-GFP staining in lumbar motor neurons but no control is included to allow comparison to the level of background staining. Images must be provided of GFP staining in the control sections taken with the same settings as for the PR mice.

Response:

We thank Reviewer 2 for this suggestion. We toned down the interpretation of synaptic transmission in “Results” and “Discussion” sections. In addition, the staining of GFP for control mice were performed and the data were added in the revised manuscript (**Supplementary Fig. 1d**). No aggregates in the lumbar motor neurons in control mice were observed.

Reviewer #3 (Remarks to the Author):

In the revised manuscript about a new C9ORF72-ALS/FTD transgenic mouse model expressing GFP-PR28 in neurons, the data quality has been improved overall. However, even not considering the caveats of the short repeat length, there is still limited novel mechanistic insight from this work. Especially the link with TDP-43 aggregation is now proved to be negative, a concern raised by several reviewers. But there are still many other pathological features related to C9 reported by many groups, including nucleocytoplasmic trafficking defects, autophagy dysfunction, ribosome dysfunction, stress granule, DNA damage, heterochromatin anomalies (recently reported in the AAV-PR50 mice). It is very important to characterize this mouse model more carefully and thoroughly and correlate these pathological features with behavior tests. This can help get a better understanding of the possible

underlying mechanisms leading to the phenotypic defects.

In addition, there are several issues with the RNA-seq analysis:

First of all, what's the cutoff of gene expression differences between two groups? Besides *P* values, the fold change threshold (such as >1.5 or 2 fold difference) also need to be taken into consideration.

In Fig. S5e, it shows there are total 267 gene changes in spinal cord. But in the Venn diagram in S5g, there are only 52 in spinal cord. Why?

In Fig.S4d, what are “interacted genes”?

When comparing the overlaps of the gene changes, including cerebellum vs spinal cord and cerebellum 2-month vs 5-month, the upregulated and downregulated genes need to be compared separately. It is possible the overlapped gene expression changes actually have opposite directions.

An earlier time point of spinal cord RNA-seq should be included and compare with cerebellum. As this manuscript focuses on motor dysfunction, it is important to test whether the synaptic transmission-related genes are also altered in spinal cord neurons.

A list of all the gene expression changes in each tissue and time point should be included in a supplementary table, which contains gene name, ID, *q* values and fold differences, et al.

Response:

1. We thank Reviewer 3 for all suggestions. As for the mechanism, although the endoplasmic reticulum (ER) stress was not enriched in the GO pathway analyses from our RNA-seq analyses, the major genes *Chac1* and *Atf5* that reflect ER-stress were both upregulated in the cerebellum of 2 months and 5 months old heterozygous mice (**Supplementary Fig. 4c, Supplementary Data 1**), suggesting that ER-stress is an early event in poly-PR expressing neurons. The upregulation of ER-stress related genes were also identified in our poly-PR expressed cultured cells, primary cortical neurons and the cerebellum of poly-PR transgenic mice (Neurosci Bull, accepted manuscript; and data not shown), which is consistent with previous published data by other groups (Kramer et al., 2018; Zhang et al., 2019). Thus, ER-stress may be a common pathological mechanism implicated in poly-PR expressing neurons. We discussed this issue in “Discussion” in our revised manuscript.
2. In data analysis, the genes with adjusted *P* values < 0.05 and $|\log_2(\text{fold change})| > 0.2$ were thought as differentially expressed. We are not the first to take the *P* values only for analysis, several groups in C9 or neurodegenerative disease research field have reported that differentially expressed genes with *P* values < 0.05 or 0.01 were selected for analyses (Kramer et al., 2018; Lin et al., 2018; Litvinchuk et al., 2018; White et al., 2018; Zhang et al., 2019). In our study, the genes with adjusted *P* values < 0.05 and $|\log_2(\text{fold change})| > 0.2$ were selected for GO pathway analyses. The representative genes from RNA-seq were further examined using qPCR analyses (Fig. 6e), which showed consistent results using these two analyses. But we do agree with Reviewer 3, a combination analysis with fold change and *P* value is a better way if there are enough genes matched both for GO pathway analyses.
3. In Fig. S5e and S5g, the numbers of gene are typo errors. We apologize for our neglect and thank Reviewer 3 much for carefulness.

4. In Fig. S4d, we did not interpret it well. “interacted genes” should be “overlapped genes”. We corrected it and here thank Reviewer 3 again for carefulness.
5. We took the advice that the upregulated and downregulated genes in the cerebellum and spinal cord of heterozygous mice were analyzed separately (**Supplementary Fig. 5g**).
6. As for the data of an earlier time point of spinal cord RNA-seq, only a few of differentially expressed genes were identified in the spinal cord, and no synaptic transmission-related genes were found in 6-month-old hetero-mice. We suppose that there should be no changes in mice at 2-month-old.
7. A list of all the genes that were changed is attached as supplementary data and the FASTQ raw data have been deposited in the NCBI GEO database.

Finally, we thank all Reviewers again for their careful review and detailed suggestions that are of help for improving our manuscript.

Sincerely,

Guanghai Wang

Reference:

- Kramer, N.J., Haney, M.S., Morgens, D.W., Jovicic, A., Couthouis, J., Li, A., Ousey, J., Ma, R., Bieri, G., Tsui, C.K., *et al.* (2018). CRISPR-Cas9 screens in human cells and primary neurons identify modifiers of C9ORF72 dipeptide-repeat-protein toxicity. *Nat Genet* 50, 603-612.
- Lin, Y.T., Seo, J., Gao, F., Feldman, H.M., Wen, H.L., Penney, J., Cam, H.P., Gjoneska, E., Raja, W.K., Cheng, J., *et al.* (2018). APOE4 Causes Widespread Molecular and Cellular Alterations Associated with Alzheimer's Disease Phenotypes in Human iPSC-Derived Brain Cell Types. *Neuron* 98, 1141-1154 e1147.
- Litvinchuk, A., Wan, Y.W., Swartzlander, D.B., Chen, F., Cole, A., Propson, N.E., Wang, Q., Zhang, B., Liu, Z., and Zheng, H. (2018). Complement C3aR Inactivation Attenuates Tau Pathology and Reverses an Immune Network Deregulated in Tauopathy Models and Alzheimer's Disease. *Neuron* 100, 1337-1353 e1335.
- White, M.A., Kim, E., Duffy, A., Adalbert, R., Phillips, B.U., Peters, O.M., Stephenson, J., Yang, S., Massenzio, F., Lin, Z., *et al.* (2018). TDP-43 gains function due to perturbed autoregulation in a Tardbp knock-in mouse model of ALS-FTD. *Nat Neurosci* 21, 552-563.
- Zhang, Y.J., Guo, L., Gonzales, P.K., Gendron, T.F., Wu, Y., Jansen-West, K., O'Raw, A.D., Pickles, S.R., Prudencio, M., Carlomagno, Y., *et al.* (2019). Heterochromatin anomalies and double-stranded RNA accumulation underlie C9orf72 poly(PR) toxicity. *Science* 363.

REVIEWERS' COMMENTS:

Reviewer #2 (Remarks to the Author):

The authors have addressed my additional concerns on the data presented.

Reviewer #3 (Remarks to the Author):

The authors addressed most questions. However, the authors didn't do any further experiments to explore alternative mechanisms. In particular, another PR mice work (ref 43) reported nucleocytoplasmic transport defects, a pathway that has also been implicated in many other previous studies. The authors should test in their model (IF of RanGAP and NPC proteins) at different time points, which is a very straight forward experiment. This is very important information for readers. The authors mentioned ER stress and claimed this is consistent with the published model (ref 43), which is not mentioned in the paper. This is very misleading.

In addition, the supplementary data have no annotation. It is not clear which file contains which data set.

We are returning the revised manuscript entitled “Motor dysfunction and neurodegeneration in a C9orf72 mouse line expressing poly-PR” (NCOMMS-18-28985C). We thank you and Reviewers for all the constructive and detailed suggestions that are of help for us to improve our manuscript. In the revised manuscript, we cited the papers that described ER stress in C9 patient samples (ref 44, Nat Neurosci, 2015) and in polyGA infected primary neurons (ref 45, Acta Neuropathol, 2014).

The followings are our point-to-point responses to the concerns raised by Reviewers.

Reviewer #2 (Remarks to the Author):

The authors have addressed my additional concerns on the data presented.

Response:

We thank Reviewer 2 for positive comment.

Reviewer #3 (Remarks to the Author):

The authors addressed most questions. However, the authors didn't do any further experiments to explore alternative mechanisms. In particular, another PR mice work (ref 43) reported nucleocytoplasmic transport defects, a pathway that has also been implicated in many other previous studies. The authors should test in their model (IF of RanGAP and NPC proteins) at different time points, which is a very straight forward experiment. This is very important information for readers. The authors mentioned ER stress and claimed this is consistent with the published model (ref 43), which is not mentioned in the paper. This is very misleading.

In addition, the supplementary data have no annotation. It is not clear which file contains which data set.

Response:

We thank Reviewer 3 for these suggestions. Dr. Petrucelli group (Ref 43 that was mentioned by reviewer 3, Science, 2019) did not mention ER stress, they described that “unfolded protein binding” and “protein folding” were major terms from their RNA-seq data. As the unfolded protein response often reflects ER-stress, and ER-stress-related genes (Atf5, Ddit3) were highly upregulated in their supplementary data, we claimed that ER-stress also occurs in PR50 mice. Because Dr. Petrucelli's study did not directly mention ER stress in text, to avoid misunderstanding, we cited the papers that described ER stress in C9 patient samples (ref 44, Nat Neurosci, 2015) and in polyGA infected primary neurons (ref 45, Acta Neuropathol, 2014).

Following the suggestions by Reviewer 3, we annotated each CSV file and also labeled in the head line in each file in Supplementary Information.

Finally, I would like to thank you again for considering our manuscript. I also thank all Reviewers for their careful review and detailed suggestions that are of much help for improving our manuscript.

Best regards,

Guanghai